# Arabidopsis α-Aurora kinase plays a role in cytokinesis through regulating MAP65-3 association with microtubules at phragmoplast midzone

Xingguang Deng [1,3] ✉, Yu Xiao[1,3], Xiaoya Tang[1], Bo Liu [2] ✉ & Honghui Lin [1] ✉

The α-Aurora kinase is a crucial regulator of spindle microtubule organization during mitosis in plants. Here, we report a post-mitotic role for α-Aurora in reorganizing the phragmoplast microtubule array. In *Arabidopsis thaliana*, α-Aurora relocated from spindle poles to the phragmoplast midzone, where it interacted with the microtubule cross-linker MAP65-3. In a hypomorphic α-Aurora mutant, MAP65-3 was detected on spindle microtubules, followed by a diffuse association pattern across the phragmoplast midzone. Simultaneously, phragmoplast microtubules remained belatedly in a solid disk array before transitioning to a ring shape. Microtubules at the leading edge of the matured phragmoplast were often disengaged, accompanied by conspicuous retentions of MAP65-3 at the phragmoplast interior edge. Specifically, α-Aurora phosphorylated two residues towards the C-terminus of MAP65-3. Mutation of these residues to alanines resulted in an increased association of MAP65-3 with microtubules within the phragmoplast. Consequently, the expansion of the phragmoplast was notably slower compared to wild-type cells or cells expressing a phospho-mimetic variant of MAP65-3. Moreover, mimicking phosphorylation reinstated disrupted MAP65-3 behaviors in plants with compromised α-Aurora function. Overall, our findings reveal a mechanism in which α-Aurora facilitates cytokinesis progression through phosphorylation-dependent restriction of MAP65-3 associating with microtubules at the phragmoplast midzone.

Cytokinesis is the final step of cell division that partitions the duplicated chromosomes and cytoplasm into two daughter cells. Although mitosis is highly conserved across eukaryotes, the mechanisms of cytokinesis utilized show remarkable diversity. In fungi and animals, cytokinesis relies on the assembly of the actin filaments into a contractile ring, which subsequently constricts the cell membrane inward, forming a cleavage furrow that separates the daughter cells[1,2]. In plant

cells, cytokinesis is achieved in a reverse manner via the formation of a new cell plate that expands from the center toward the periphery of the cell[3].

The phragmoplast, composed of microtubules, actin filaments, and endomembrane compartments, is responsible for fusing the cell plate in plants. In the phragmoplast, microtubules organize into two mirrored sets of anti-parallel arrays, interdigitating their plus ends at

[1]Key Laboratory of Bio-resource and Eco-environment of Ministry of Education, College of Life Sciences, State Key Laboratory of Hydraulics and Mountain River Engineering, Sichuan University, Chengdu, China. [2]Department of Plant Biology, College of Biological Sciences, University of California, Davis, CA, USA. [3]These authors contributed equally: Xingguang Deng, Yu Xiao. ✉e-mail: xgdeng@scu.edu.cn; bliu@ucdavis.edu; hhlin@scu.edu.cn

the equatorial zone of the apparatus[4,5]. Golgi-derived vesicles carrying cell plate materials are transported along microtubules towards their plus ends and then fuse to create a new cross wall at the phragmoplast midzone[4]. During cytokinesis, the phragmoplast expands centrifugally from the interiors to the periphery of the cell, resulting in the inside-out growth of a cell plate[5,6].

The expansion of the phragmoplast requires microtubule turnover, whereby new microtubules assemble at the expanding edge while older microtubules disassemble at the center[6,7]. These events are supposedly regulated by proteins in the 65-kD microtubule-associated protein (MAP65) family. MAP65 proteins can bundle microtubules by forming homodimers[8]. Their homologs, including the fungal protein Ase1 (Anaphase Spindle Elongation1) and mammalian protein PRC1 (Protein Regulator of Cytokinesis1), act as antiparallel microtubule bridges that are required to establish and maintain the central spindle[9]. In *Arabidopsis thaliana*, there are nine MAP65 isoforms comprising an N-terminal dimerization region and a C-terminal microtubule-binding domain[10–12]. Although MAP65 proteins collectively are thought to function at the phragmoplast midzone, inactivation of *MAP65-1* or *MAP65-2* gene in *A. thaliana* does not lead to obvious phenotypes in cell division[13]. In contrast, null *map65-3* mutations frequently cause cytokinesis failure, resulting in cell wall stubs and the formation of giant cells with multiple nuclei[14–16]. These observations suggest the MAP65-3 isoform possesses some unique functions in the phragmoplast that are not shared by other isoforms under native conditions. In particular, MAP65-3 has been recognized as a cytokinesis-specific MAP65 isoform that primarily interdigitates phragmoplast microtubules[14,16]. Towards later stages of mitosis, MAP65-3 emerges in the middle segments of the anaphase spindle and subsequently concentrates at the midzone, where it stabilizes the phragmoplast configuration by cross-linking antiparallel microtubules[16]. The release of antiparallel microtubule overlaps in the phragmoplast is regulated by the phosphorylation of MAP65 isoforms[17]. Phosphorylation triggers MAP65 proteins detachment from microtubules, facilitating microtubule turnover and phragmoplast expansion[4]. The mitogen-activated protein kinase (MAPK) cascade is the best-understood signaling pathway involved in this phosphorylation[18–22]. However, mimicking MAP65-1 phosphorylation by MAPK only slightly decreases its ability to bind and bundle microtubules[11], suggesting that other kinases may also be involved in regulating MAP65 proteins via phosphorylation during cytokinesis.

The evolutionarily conserved Aurora kinases are key regulators for cell division across eukaryotes. Vertebrate Aurora can be subdivided into two functionally divergent groups consisting of Aurora-A and Aurora-B/C[23]. Aurora-A associates with the centrosome during early mitosis and regulates spindle formation, while Aurora-B is part of the chromosomal passenger complex that controls chromosome biorientation and segregation[24,25]. The functional specification of Aurora kinases depends on their binding partners. In contrast to fungi and animals, very little has been learned regarding Aurora function during cell division in plants. Like vertebrates, plant Aurora kinases are classified into the spindle associated α-group (including AUR1 and AUR2) and the centromere localized β-group (AUR3)[26]. The α-Aurora localizes predominantly to spindle poles and plays a pivotal role in sustaining spindle bipolarity[26,27]. Unlike its mammalian counterpart Aurora-A, α-Aurora activation and localization depend on TPXL3 rather than the canonical TPX2 in *A. thaliana*[27,28]. Intriguingly, after cytokinesis onset, α-Aurora transports from spindle poles to the phragmoplast midzone[26,29]. In addition, inhibiting or knocking down α-Aurora in *A. thaliana* results in a defective cytokinesis phenotype associated with irregular cell plate orientation[29]. These findings suggest that α-Aurora may play a role in plant cytokinesis, although its binding partners and substrates at the phragmoplast midzone remain to be identified.

The striking localization of α-Aurora to the phragmoplast midzone led us to investigate whether compromising α-Aurora function specifically impacts cytokinesis in plants. Recently, MAP65-1 has been thought to be a substrate of α-Aurora kinase in plants[30], but MAP65-1 exhibits distinct localization from α-Aurora and probably does not function at the phragmoplast midzone where microtubules interdigitate[31]. Through this work, we discover that the microtubule cross-linker MAP65-3 is an interacting partner and downstream target of the α-Aurora kinase at the phragmoplast midzone in *A. thaliana*. Our findings reveal that α-Aurora phosphorylation negatively regulates MAP65-3's ability to associate with microtubules and is required for proper phragmoplast expansion during cytokinesis.

## Results

### α-Aurora localizes at the phragmoplast midzone during cytokinesis

Our recent studies indicated an N-terminal GFP-AUR1 fusion expressed under its native promotor fully rescued the dwarf and bushy growth phenotype of the α-Aurora double knock-down mutant (*aur1 aur2*)[28]. Here, we used this transgenic line to examine the dual localization of AUR1 and microtubules via immunofluorescence experiments. As shown in Supplementary Fig. 1, around nuclear envelope breakdown, GFP-AUR1 was detected prominently at the pro-spindle polar caps. At metaphase when chromosomes aligned at the equatorial plate, GFP-AUR1 signal was present on the spindle apparatus decorating kinetochore fibers. In early anaphase, AUR1 is highly concentrated on the shortened kinetochore fibers at spindle poles. During telophase, AUR1 was detected at the distal ends of the emerging phragmoplast microtubules. When the phragmoplast began centrifugal expansion, GFP-AUR1 concentrated at the phragmoplast midzone, where microtubule density was lowest based on immunofluorescence. At this stage, AUR1 was undetectable at spindle poles. When the phragmoplast disk nearly reached the cell periphery, AUR1 remained at the equator even after microtubule disassembly at the phragmoplast center. These results reveal that in addition to associating with spindle microtubules, Arabidopsis α-Aurora also targets the phragmoplast midzone.

### α-Aurora facilitates the expansion of the phragmoplast microtubule array

To determine if α-Aurora functions at the phragmoplast midzone, we assessed phragmoplast dynamics in the *aur1 aur2* double mutant. To do so, we expressed a GFP-TUB6 fusion to mark microtubules and performed live-cell imaging from late telophase to the end of cytokinesis in wild-type and mutant cells. Following assembly of the bipolar phragmoplast microtubule assay at the cell center, the solid array expanded toward the cell periphery as microtubules disappeared from interior areas. In wild-type cells, the process of cytokinesis, defined from initiation of the phragmoplast to its full disassembly, required approximately 20 minutes after phragmoplast expansion had commenced (Fig. 1a, Supplementary Movie 1). However, in *aur1 aur2* cells with similar size, phragmoplast microtubule arrays persisted for a significantly longer time of 35 min during cytokinesis (Fig. 1b, Supplementary Movie 2). Kymograph analysis along the expansion axis showed that fluorescence spread more slowly in *aur1 aur2* cells than in controls (Fig. 1c). The mean velocities of phragmoplast expansion were $0.640 \pm 0.041\,\mu m/min$ ($n = 12$ cells) in wild-type cells and $0.392 \pm 0.033\,\mu m/min$ ($n = 12$ cells) in *aur1 aur2* cells (Fig. 1d), indicating reduced expansion rate of the phragmoplast in mutant cells prolonged cytokinesis process.

At late cytokinesis when the phragmoplast became ring-shaped, microtubule bundles in *aur1 aur2* cells were loosely packaged and disorganized. Besides a wider gap at the midzone, the overall phragmoplast length (i.e., the distance between its two distal ends) appeared longer than its tightly packed solid array at early cytokinesis (Fig. 1b). In *aur1 aur2* cells, the phragmoplast length increased approximately

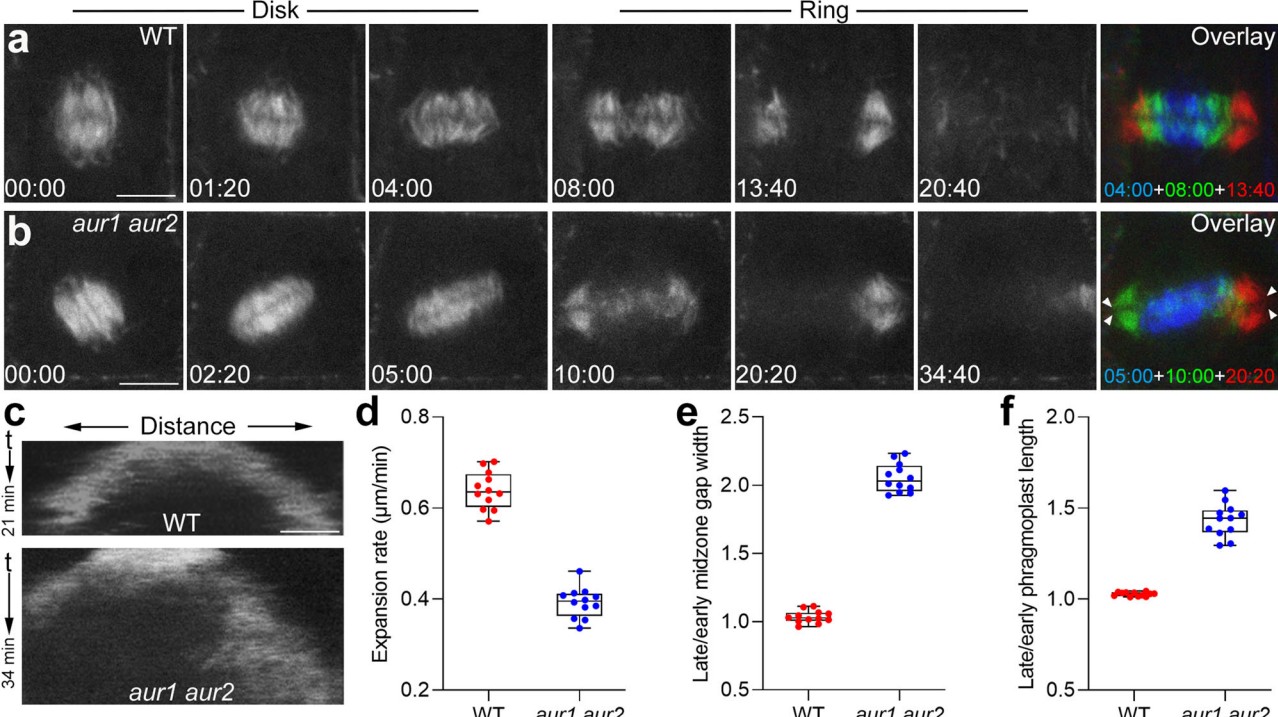

**Fig. 1 | Inhibition of α-Aurora affects phragmoplast expansion. a** Live imaging of wild-type (WT) cells expressing GFP-TUB6. The overlay shows the merges of three representative time points of phragmoplast expansion using different colors and indicates no alterations in orientation or midzone width of the phragmoplast between early and late cytokinesis. Bars, 5 μm. Representative images are acquired from Supplementary Movie 1. **b** Live imaging of *aur1 aur2* cells expressing GFP-TUB6. The overlay shows the misalignment of phragmoplast orientation between early and late cytokinesis. Wider phragmoplast midzones are indicated by arrowheads at late cytokinesis. Bars, 5 μm. Images are acquired from Supplementary

Movie 2. **c** Kymographs of the microtubule signal along the expansion axis in WT and *aur1 aur2* cells. The x-coordinate renders the distance, and the y-coordinate indicates the time. Bars, 2.5 μm. **d–f** Quantitative assessment of the phragmoplast expansion velocity (**d**), the ratio of late and early phragmoplast midzone width (**e**), and the ratio of late and early phragmoplast length (**f**) in WT and *aur1 aur2* cells (*n* = 12 cells). Data are presented as box-and-whisker plots with individual points, showing the interquartile range (box), the median (horizontal line), and minimum and maximum values (whiskers).

1.5-fold (Fig. 1f), and the midzone gap width increased over 2-fold from the beginning to the ending of cytokinesis (Fig. 1e). In contrast, as the phragmoplast expanded, its length and gap width remained largely unchanged in wild-type cells. (Fig. 1e, f). Besides increased length, the phragmoplast exhibited extensive rotational motions in *aur1 aur2* plants but remained nearly horizontal in wild-type plants (Fig. 1a, b). Overall, these observations imply that α-Aurora is involved in regulating phragmoplast dynamics in *A. thaliana*.

### α-Aurora interacts with the N-terminal of MAP65-3
The specific distribution of α-Aurora during cytokinesis is reminiscent of MAP65-3, an effective crosslinker of antiparallel microtubules essential for phragmoplast integrity. Moreover, the abnormal ring-shaped phragmoplast with wider midzones in *aur1 aur2* cells resembles MAP65-3 impairment in *A. thaliana*. These similarities prompt us to explore potential interactions between the two proteins. Yeast two-hybrid (Y2H) and bimolecular fluorescence complementation (BiFC) assays showed AUR1 directly interacted with MAP65-3 (Supplementary Fig. 2). BiFC generated a fluorescence signal in nuclei where AUR1-MAP65-3 complexes localized (Supplementary Fig. 2b). This nuclear localization also supported the interaction since MAP65-3 normally decorates cortical microtubules. As AUR1 localized to interphase nuclei, the interaction recruited MAP65-3 from microtubules to nuclei. Intriguingly, we found that no interaction took place between AUR3 and MAP65-3 through both Y2H and BIFC assays (Supplementary Fig. 2), demonstrating MAP65-3 specifically associates with Arabidopsis α- but not β-group Aurora kinase.

To learn which domain in MAP65-3 is responsible for binding AUR1, we made truncated MAP65-3 variants for Y2H assays (Fig. 2a). We first separated MAP65-3 into the N-terminal dimerization fragment and the C-terminal microtubule-binding region. We found that the dimerization domain in MAP65-3 was sufficient for interacting with AUR1, no interaction took place at the microtubule-binding part (Fig. 2b). Further dissecting the N-terminus of MAP65-3 abolished AUR1 interaction, indicating that the interaction requires the entire N-terminus. Physical interaction with the N-terminal MAP65-3 fragment was further confirmed by pull-down assays using GST-tagged AUR1 and His-fused MAP65-3 truncations (Fig. 2c).

### Disrupted microtubule association and turnover of MAP65-3 in *aur1 aur2* plants
The cytokinesis phenotype in the *aur1 aur2* mutant, as well as the interaction, prompt us to ask if the microtubule associating patterns of MAP65-3 are affected in the absence of α-Aurora. To test this, a functional MAP65-3-GFP fusion expressed by its native promoter was transformed into *aur1 aur2* and *map65-3* plants, and dual localization analysis was carried out to show the relationship between MAP65-3 and microtubules. In control cells, MAP65-3 fluorescence was barely detected at metaphase and early anaphase. Whereas in *aur1 aur2* cells, MAP65-3 is particularly associated with spindle microtubule bundles at both stages (Fig. 3a, b). Upon completing chromosome segregation, MAP65-3 fluorescence appeared particularly on the central spindle mid-region in control cells. However, at the same stage in *aur1 aur2* cells, MAP65-3 signal appeared more broadly distributed along central spindle microtubules instead of the narrowed concentration at the

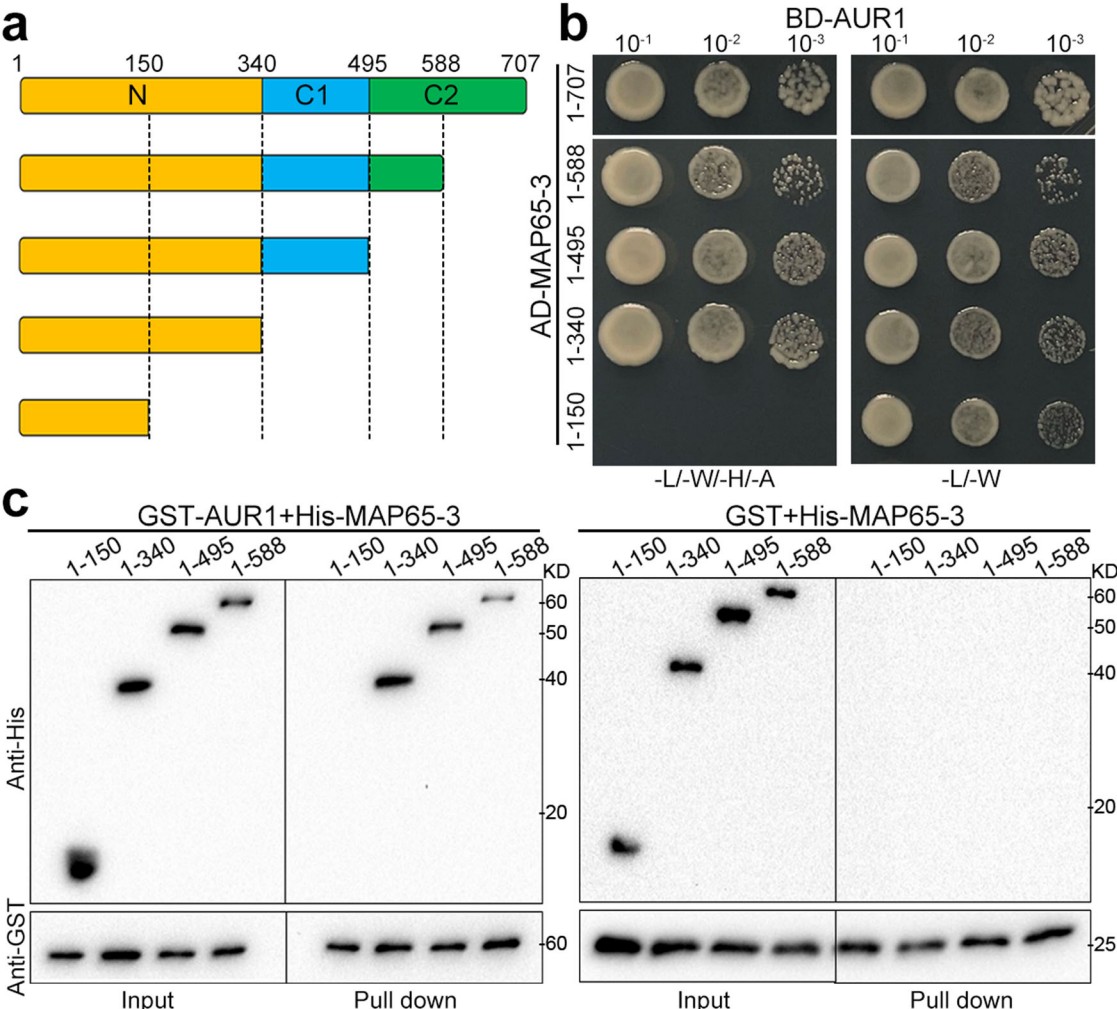

**Fig. 2 | α-Aurora interacts with MAP65-3. a** Schematic representation of full-length and truncated versions of MAP65-3 used to map α-Aurora binding regions. **b** Yeast two-hybrid assay to examine interactions between AUR1 and truncated MAP65-3 variants. The yeast cultures were spotted on interaction-selective (-L/-W/-H/-A, left column) and vector-selective (-L/-W, right column) media and

photographed after incubation at 30 °C for 2 days. **c** In vitro pull-down assays of recombinant His fusions of truncated MAP65-3 variants with GST-AUR1 and GST control immobilized on glutathione beads. The experiments were repeated three times with similar results.

midzone observed in control cells (Fig. 3c). When the central spindle was replaced by an early phragmoplast, MAP65-3 localized intensely at the phragmoplast midzone in control cells. However, in *aur1 aur2* cells, MAP65-3 signal became less restrained, with punctate signals along phragmoplast microtubules (Fig. 3d).

We then analyzed the distribution patterns of MAP65-3 with microtubules in the central spindle and the early phragmoplast by quantitating fluorescence intensities. At telophase, MAP65-3 fluorescence almost completely overlapped central spindle microtubules in *aur1 aur2* plants (Fig. 3f). In control plants, MAP65-3 signal coincided only with interzonal microtubules at the central spindle midzone (Fig. 3e). At early cytokinesis, MAP65-3 occupied the phragmoplast midzone where tubulin signal dropped sharply in control cells (Fig. 3g). However, in *aur1 aur2* cells, the average MAP65-3 fluorescence width was wider than the phragmoplast midzone (Fig. 3h). These findings indicate that disruption of α-Aurora function results in MAP65-3 displaying an earlier and wider microtubule decoration patterns as cytokinesis commenced.

Conversely, we ask whether impairing MAP65-3 could impact α-Aurora localization. We compared GFP-AUR1 signal in complemented control and *map65-3* plants. GFP-AUR1 localized comparably to mitotic spindles and cell plates in both backgrounds (Supplementary Fig. 3,

Supplementary Movie 3-4). Therefore, we conclude α-Aurora likely achieves its localization independently of MAP65-3.

To further investigate how α-Aurora affects MAP65-3 associating with phragmoplast microtubules dynamically. Fluorescence recovery after photobleaching (FRAP) was applied to measure MAP65-3-GFP turnover during cytokinesis (Fig. 4a). In control cells, MAP65-3 fluorescence clearly recovered during the observation period (approximately within 120 s, Supplementary Movie 5). By contrast, recovery of the fluorescence in *aur1 aur2* cells was largely delayed (Supplementary Movie 6). Quantitation showed the half-life ($t_{1/2}$) of fluorescence recovery was 75.5 s in control cells, whereas it was extended to 263.5 s in *aur1 aur2* cells (Fig. 4b). The slower MAP65-3 turnover in *aur1 aur2* cells was also observed at late cytokinesis stages (Supplementary Fig. 4).

The delayed MAP65-3 turnover was further evaluated by comparison of its localization at late cytokinesis. In control cells, MAP65-3 disassembled from the interior region as the phragmoplast transitioned from a disk shape to a ring shape. At the same stage in *aur1 aur2* cells, MAP65-3 retained a substantial signal at the dismantling center (Fig. 4c). When the phragmoplast became fully ring-shaped, distributions of MAP65-3 in control and *aur1 aur2* cells were quite different. In control cells, MAP65-3 intensity was maximal inside phragmoplast

microtubule arrays, and its intensity at leading and trailing edges was indistinguishable. However, in *aur1 aur2* cells, MAP65-3 preferentially accumulated at the matured phragmoplast's trailing edge (Fig. 4c). In 3-D projections of a mature phragmoplast, MAP65-3 was detected as a ring-like structure in control cells. While in *aur1 aur2* cells, considerable MAP65-3 signal still persisted at the center of the ring-shaped phragmoplast (Fig. 4d, Supplementary Movie 7-8). Live-cell imaging of MAP65-3-GFP after having mCherry-TUB6 co-expressed showed that MAP65-3 exhibits delayed movement from interiors to leading edges of the phragmoplast during cytokinesis in *aur1 aur2* plants (Fig. 4e, Supplementary Movie 9-10). Some proteins thought to interact with MAP65-3, including BUB3.1, Kinesin-12A and TRS120, also exhibited consistent localization dynamics similar to that of MAP65-3 in *aur1 aur2* plants (Supplementary Fig. 5). Thus, we propose that the delayed MAP65-3 turnover may cause the unzipped leading edges of maturing phragmoplasts in *aur1 aur2* plants.

We further investigated whether cell plate assembly is impacted by altered MAP65-3 behaviors in *aur1 aur2* plants. Using the membrane fusion SNARE protein KNOLLE as a cell plate marker, we selected cells with two separated nuclei and a mature phragmoplast. In *aur1 aur2* cells, mature cell plates were detected (Fig. 5a, b), though the KNOLLE signal appeared slightly thicker than in controls (Fig. 5c), suggesting minor defects in membrane trafficking or fusion. Intriguingly, live-cell imaging revealed some cells in the *aur1 aur2* mutant formed branched cell plates, as detected by the FM4-64 outlining normal cell shapes (Fig. 5d, e). These aberrant cell plates consistently presented with MAP65-3 remaining at branch initiation points or accumulating throughout the entire branch region (Fig. 5f, Supplementary Movie 11). This association implies that MAP65-3 failing to withdraw from the dismantling area in a timely manner could be linked to abnormal cell plate development.

### MAP65-3 is a substrate of α-Aurora
Although Aurora-dependent C-terminal phosphorylation of MAP65-1 has been reported in Arabidopsis and Medicago[30,32], whether α-Aurora phosphorylates MAP65-3, which has a distinct C-terminus from MAP65-1, remains unknown. As the above results reveal the interaction between AUR1 and MAP65-3, we then investigate whether MAP65-3 is an α-Aurora substrate. In vitro kinase assays showed that recombinant GST-AUR1 phosphorylated His-MAP65-3 (Fig. 6b). Mass spectrometry of in vitro phosphorylated MAP65-3 peptides identified two phosphorylated residues (S528 and S570) in the MAP65-3 C-terminus (Fig. 6a, Supplementary Data 1). To confirm if these sites were α-Aurora targets, we performed in vitro kinase assays using MAP65-3 phospho-mutants with either or both residues substituted to Ala. Mutation with one substitution (S528A or S570A) in MAP65-3 showed unnoticeable effects on AUR1-dependent phosphorylation from its unmodified form. However, the double mutant derivative of MAP65-3 (S528A & S570A, abbreviated as MAP65-3$^{AA}$) was barely phosphorylated by recombinant AUR1 (Fig. 6b). These results demonstrate MAP65-3 is a bona fide substrate of α-Aurora.

### Phosphorylation by α-Aurora restricts MAP65-3 associating with microtubules
We further explored how α-Aurora-dependent phosphorylation regulates MAP65-3 function on a cellular level. For this purpose, various phospho-defective and phospho-mimetic mutation forms of MAP65-3 were expressed by its endogenous promoter in the *map65-3* background. The impaired root growth of *map65-3* seedlings was completely rescued by the expression of either MAP65-3 or MAP65-3$^{DD}$. Expression of MAP65-3$^{AA}$ partially but not completely restored the growth defects caused by the *map65-3* mutation (Supplementary Fig. 6).

We then examined the microtubule colocalization patterns of these phospho-isoforms when expressed in *map65-3* mutant cells.

As expected from fully rescuing the mutant seedlings, MAP65-3$^{DD}$ distributed like the unmodified protein, primarily associating with the central spindle midzone and later focusing within the phragmoplast midzone. However, MAP65-3$^{AA}$ was detected not exclusively at the midzone but also along microtubule bundles in central spindles (Fig. 6c). Intensity plots of average fluorescence along the central spindle length confirmed a much wider distribution pattern of MAP65-3$^{AA}$ than MAP65-3$^{DD}$ (Fig. 6d, e). Moreover, substantial MAP65-3$^{AA}$ fluorescence was also observed, concentrating within the disassembling interior regions of the phragmoplast (Fig. 6c).

We also compared turnover rates between MAP65-3$^{AA}$ and MAP65-3$^{DD}$ via FRAP (Fig. 6f). During cytokinesis, MAP65-3$^{DD}$ performed a comparable turnover rate to its unmodified form, with half-time values of 77.4 s (Fig. 6g, Supplementary Movie 12). However, MAP65-3$^{AA}$ had a stable appearance with low turnover rates of half-time values at 213.7 s (Fig. 6g, Supplementary Movie 13). We further examined the turnover of MAP65-3 phospho-variants in tobacco epidermal cells through transient expression assays (Supplementary Fig. 7). FRAP analysis showed MAP65-3$^{AA}$ detached from microtubules more slowly compared to the unmodified MAP65-3, exhibiting delayed fluorescence recovery. In contrast, MAP65-3$^{DD}$ turnover was enhanced slightly, with a little rapid signal recovery.

Finally, the dynamics of MAP65-3 phospho-variants were evaluated by time-lapse imaging in *map65-3* plants. Plants co-expressed the microtubule marker mCherry-TUB6 to assess phragmoplast expansion. As shown in Fig. 6h, MAP65-3$^{AA}$ initially appeared in a wider midzone region at telophase and subsequently persisted for an extended period of time in a disk-shaped structure (Supplementary Movie 14). In contrast, MAP65-3$^{DD}$ concentrated at the central spindle midzone at telophase and quickly underwent a transition from a disk shape to a ring shape (Supplementary Movie 15). We quantified phragmoplast expansion rates for each MAP65-3 variant (Fig. 6i). MAP65-3$^{DD}$ lines showed a mean expansion velocity of $0.644 \pm 0.040$ μm/min ($n = 10$ cells), while MAP65-3$^{AA}$ lines expanded at a rate of $0.410 \pm 0.038$ μm/min ($n = 10$ cells). Colocalization analysis of MAP65-3 and FM4-64 stained cell plates in those lines showed MAP65-3$^{AA}$ persisted substantially along a mature cell plate, while MAP65-3$^{DD}$ exclusively accumulated at the outer edge of the matured cell plate (Fig. 6j).

Overall, the abnormal microtubule association and dynamics of MAP65-3 resulting from the phospho-defective mutation were consistent with observations in the *aur1 aur2* double mutant.

### Phosphorylation mimicry restores MAP65-3 behaviors in *aur1 aur2* plants
To further explore if α-Aurora regulates MAP65-3 activity through phosphorylation *in planta*, we explored the behaviors of phospho-mimetic MAP65-3 in the α-Aurora mutant background. When expressed in *aur1 aur2* plants, which exhibited a dwarf and bushy growth phenotype due to compromised α-Aurora function, neither MAP65-3 nor MAP65-3$^{AA}$ affected this bushy growth phenotype. However, the phospho-mimetic MAP65-3$^{DD}$ showed a subtle suppression of the compact growth phenotype, making the double mutant plants look a little bigger but not in an obvious way (Fig. 7a). As observed for native MAP65-3 distribution in *aur1 aur2* cells, MAP65-3$^{AA}$ exhibited prolonged and diffuse signal at telophase, followed by retention at the phragmoplast interiors during cytokinesis (Fig. 7b). In contrast, distribution of MAP65-3$^{DD}$ in *aur1 aur2* cells was comparable to its unmodified form in control cells, which concentrated specifically at the midzone of central spindles and expanding phragmoplasts (Fig. 7c).

We then performed time-lapse imaging of MAP65-3 phospho-variants expressed in *aur1 aur2* cells. FM4-64 staining was used to visualize cell plate expansion for analyzing relationships between MAP65-3 dynamics and cytokinesis progression. Again, MAP65-3$^{AA}$

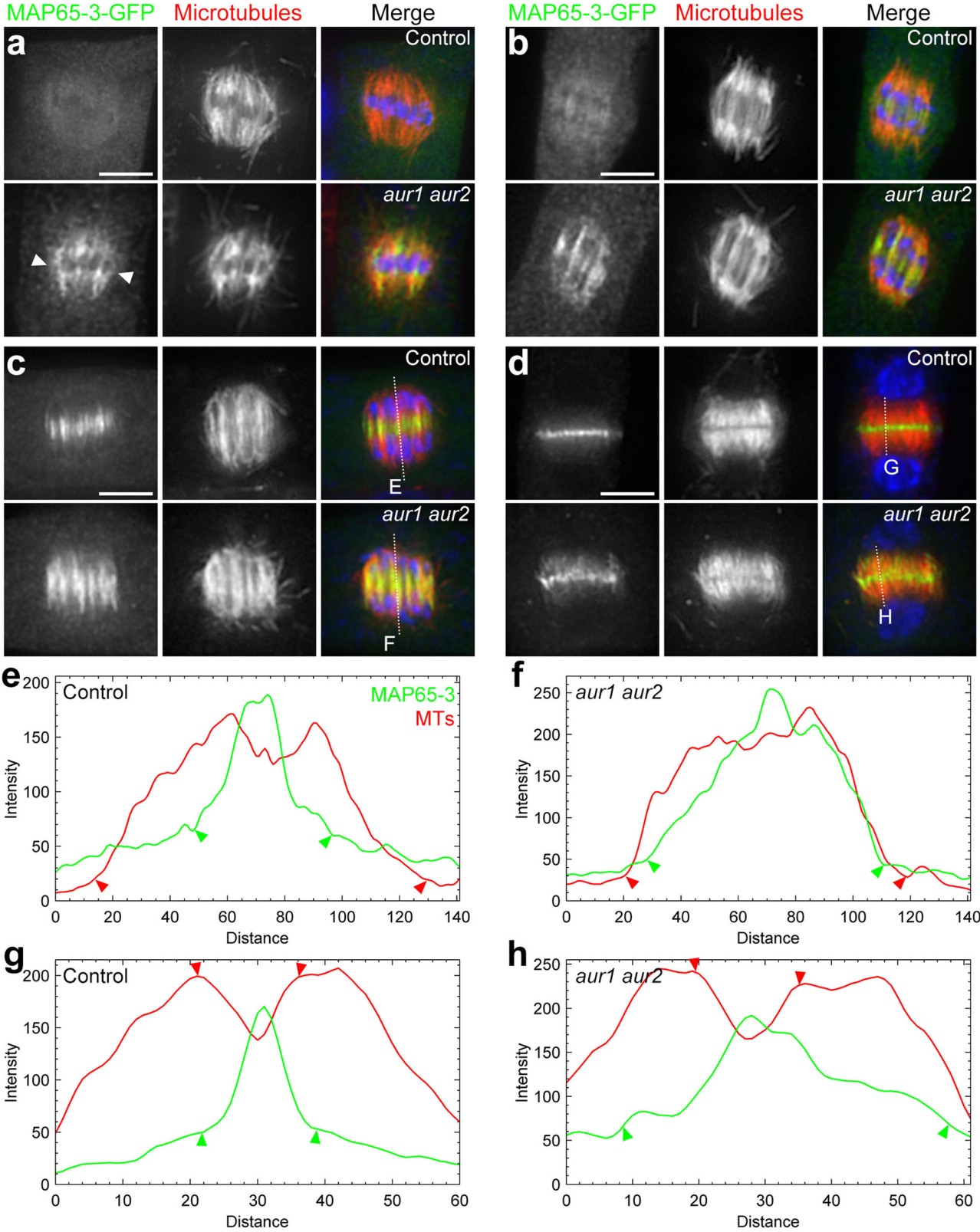

showed a prolonged disk-shaped signal, followed by persistent enrichment and retention at the inside region as the cell plate expanded and matured (Fig. 7d, Supplementary Movie 16). While MAP65-3$^{DD}$ transitioned rapidly to decorate the rim of the expanding cell plate and marked the leading edge through cell plate maturation (Fig. 7e, Supplementary Movie 17). Quantitation showed MAP65-3$^{DD}$ significantly restored the cell plate expansion rate ($0.515 \pm 0.036$ μm/min, $n = 8$ cells) compared to MAP65-3$^{AA}$ ($0.314 \pm 0.021$ μm/min, $n = 8$ cells) in *aur1 aur2* plants (Supplementary Fig. 8).

Collectively, these findings demonstrate that in *aur1 aur2* plants, mimicking α-Aurora-mediated phosphorylation of MAP65-3 via the MAP65-3$^{DD}$ variant helps restore its typical distributions and dynamics

**Fig. 3 | Inhibition of α-Aurora causes an earlier and wider distribution of MAP65-3.** **a–d** Comparative distribution of MAP65-3 in *map65-3* (control) and *aur1 aur2* backgrounds. At metaphase (**a**), or anaphase (**b**), MAP65-3 is barely detected in control cells, while the fluorescence signal decorates spindle microtubule bundles in *aur1 aur2* cells. At telophase (**c**), MAP65-3 decorates a narrow region of the central spindle in control cells, while the signal covers almost the whole central spindle in *aur1 aur2* cells. At early cytokinesis when a phragmoplast is formed (**d**), MAP65-3 becomes less concentrated in the phragmoplast midzone in *aur1 aur2*

cells when compared with control cells. All merged images have MAP65-3-GFP detected by the anti-GFP antibody in green, microtubules in red, and DNA in blue. Bars, 5 μm. **e–h** Assessment of MAP65-3 distribution patterns in control and *aur1 aur2* cells by fluorescence intensity scans of the regions indicated by dotted lines from merged images. The lowest MAP65-3 points are highlighted by green arrowheads. The edges of the central spindle microtubules and the phragmoplast midzone are illustrated by red arrowheads. Micrographs are representative of more than 100 cells from three independent lines with similar results.

within the phragmoplast. This likely contributes to rescuing the delayed expansion of the phragmoplast/cell plate seen in this mutant background where α-Aurora function is compromised.

## Discussion

The Aurora-A kinase family has long been known as an essential mitotic spindle assembly factor in eukaryotes. Here, our results demonstrate that this kinase family (named α-Aurora in *A. thaliana*) has evolved to function in phragmoplast-based cytokinesis in plants. Such a function is established through its specific interaction and phosphorylation of the MAP65 family protein MAP65-3. The α-Aurora dependent phosphorylation facilitates MAP65-3 turnover at the phragmoplast midzone and promotes the expansion of the phragmoplast. By revealing a pathway in which MAP65-3 acts as a crucial intermediary, we provide further insight into how α-Aurora signaling drives cytokinesis progression in plants.

Although the plant α-group Aurora displays functional properties typical of the mammalian Aurora-A clade, such as establishing bipolar spindles and modifying histones[26,29,33], α-Aurora localizes to the cytokinetic apparatus midzone where the cell plate forms, resembling mammalian Aurora-B[26,29]. In mammals, Aurora-B is targeted to the overlapped microtubules of the central spindle and promotes contractile ring formation during cytokinesis[25]. However, the plant β-Aurora protein does not translocate from kinetochores to the forming cell plate paralleling Aurora-B at the cell division site during cytokinesis[26,34]. Remarkably, our results, as well as previous studies, reveal that the two α-Aurora members, AUR1 and AUR2, localize to the division plane until cytokinesis ends[25]. The specific distribution of α-Aurora at the phragmoplast midzone indicates a site-specific function. In the weak loss-of-function *aur1 aur2* double mutant, the phragmoplast expands considerably slower than in wild-type. The observation of loosely packaged phragmoplast leading edges with a wider midzone in *aur1 aur2* plants highlights α-Aurora's cytokinesis-specific function at organizing microtubules during phragmoplast maturation.

Binding with different interaction partners defines spatial localization and, therefore, different functions of Aurora kinases[23]. TPX2 is one of the most well-known interaction partners that control the localization and activity of Aurora-A at mitotic spindles[24]. Our recent study illustrates that TPX2 loses this canonical function in *A. thaliana*, instead, a related TPXL3 defines the localization and mitotic functions of α-Aurora[27]. Currently, it is unclear how the cytokinesis-specific localization is achieved by α-Aurora, as TPXL3 does not localize to the phragmoplast midzone during cytokinesis. In mammals, PRC1 is an Aurora-B binding partner but not a substrate during cytokinesis[35,36]. The proper localization of Aurora-B at the central spindle midzone is dependent on PRC1[35]. However, our results show a contrary situation in *A. thaliana*. During cytokinesis, α-Aurora not only interacts with MAP65-3 but also phosphorylates it as a substrate. However, impairing MAP65-3 does not alter α-Aurora's localization at the phragmoplast midzone. This suggests that additional unidentified partners that likely have microtubule plus-end binding affinity probably target α-Aurora to the division plane in plants.

The expansion of the phragmoplast microtubule array is affected in the *aur1 aur2* double mutant, indicating microtubule turnover may be impaired. Microtubule dynamics in the phragmoplast are governed

by MAP65-3, which maintains phragmoplast structure by stabilizing microtubule overlaps at the midzone[6,37]. We clarify the inadequate lateral phragmoplast expansion in the *aur1 aur2* mutant results from compromised MAP65-3 distributing dynamics during cytokinesis. In *aur1 aur2* cells, MAP65-3 decorates a wider phragmoplast microtubule area outside the mid zone, potentially over-stabilizing the phragmoplast and restricting microtubule depolymerization internally. This likely slows phragmoplast expansion. Furthermore, MAP65-3 turnover is diminished in the *aur1 aur2* mutant, causing its predominant retention at the phragmoplast interior edge. As a result, newly polymerized microtubules from phragmoplast halves may fail to bridge at the expanding periphery in the absence of MAP65-3. This impaired establishment of interdigitated microtubules could, therefore, account for the unzipped leading edge observed in the mature phragmoplast in *aur1 aur2* plants.

Our results demonstrate that the α-Aurora kinase promotes the disassembly of MAP65-3 from microtubule overlap regions near the phragmoplast midzone by reducing its microtubule-binding activities, thereby facilitating its turnover at key stages of cytokinesis. Such a function may be established based on phosphorylating modification. In flowering plants, the microtubule-binding activity of MAP65 family proteins is regulated by MAPK-dependent phosphorylation[20,38–41]. It has been reported that phosphorylation of MAP65-1 by MAPK cascade down-regulates its microtubule bundling properties, which stimulates microtubule turnover and phragmoplast expansion in plants[21,31]. Recent studies reveal that α-Aurora interacts with and phosphorylates MAP65-1 in Arabidopsis and Medicago[30,32]. Although MAP65-1 functions redundantly with MAP65-3, it is believed that MAP65-3 plays a more critical role in phragmoplast stabilization[13,42,43]. Therefore, α-Aurora may exert its effects on cytokinesis through phosphorylating MAP65-3 instead of MAP65-1 in plants. Our results confirm MAP65-3 as a bona fide substrate of α-Aurora in *A. thaliana* and reveal that phosphorylation by α-Aurora affects MAP65-3 distribution and dynamics within phragmoplast microtubule arrays. First, the non-phosphorylatable MAP65-3[AA] decorates a wider region and turnovers more slowly on microtubules than in its unmodified form. Furthermore, MAP65-3[AA] does not properly translocate to leading edges, associating instead with maturing phragmoplast interiors. Third, expressing MAP65-3[AA] does not fully rescue deficiencies (e.g., root length and phragmoplast expansion rate) caused by *MAP65-3* loss. Finally, mimicking the phosphorylated state of MAP65-3 could restore its native manners in the absence of α-Aurora kinase. We propose that α-Aurora-mediated phosphorylation facilitates MAP65-3 detaching from the interdigitated microtubules and expedites microtubule turnovers in the phragmoplast. The phosphorylation may also impact how MAP65-3 interacts with other binding proteins besides microtubules, which could, in turn, help regulate phragmoplast organization. Thus, α-Aurora kinase signaling provides another mode of modulating MAP65 activities that could function alongside the MAPK pathway. A system of checks and balances, including α-Aurora and MAPKs, may act cooperatively or redundantly to govern MAP65-3 and ensure proper phragmoplast progression.

We identify two MAP65-3 phospho-sites targeted by α-Aurora in Arabidopsis: Ser-528 and Ser-570. Both sites are located at the extreme C-terminal (C2) region in MAP65-3. Intriguingly, the C-terminus

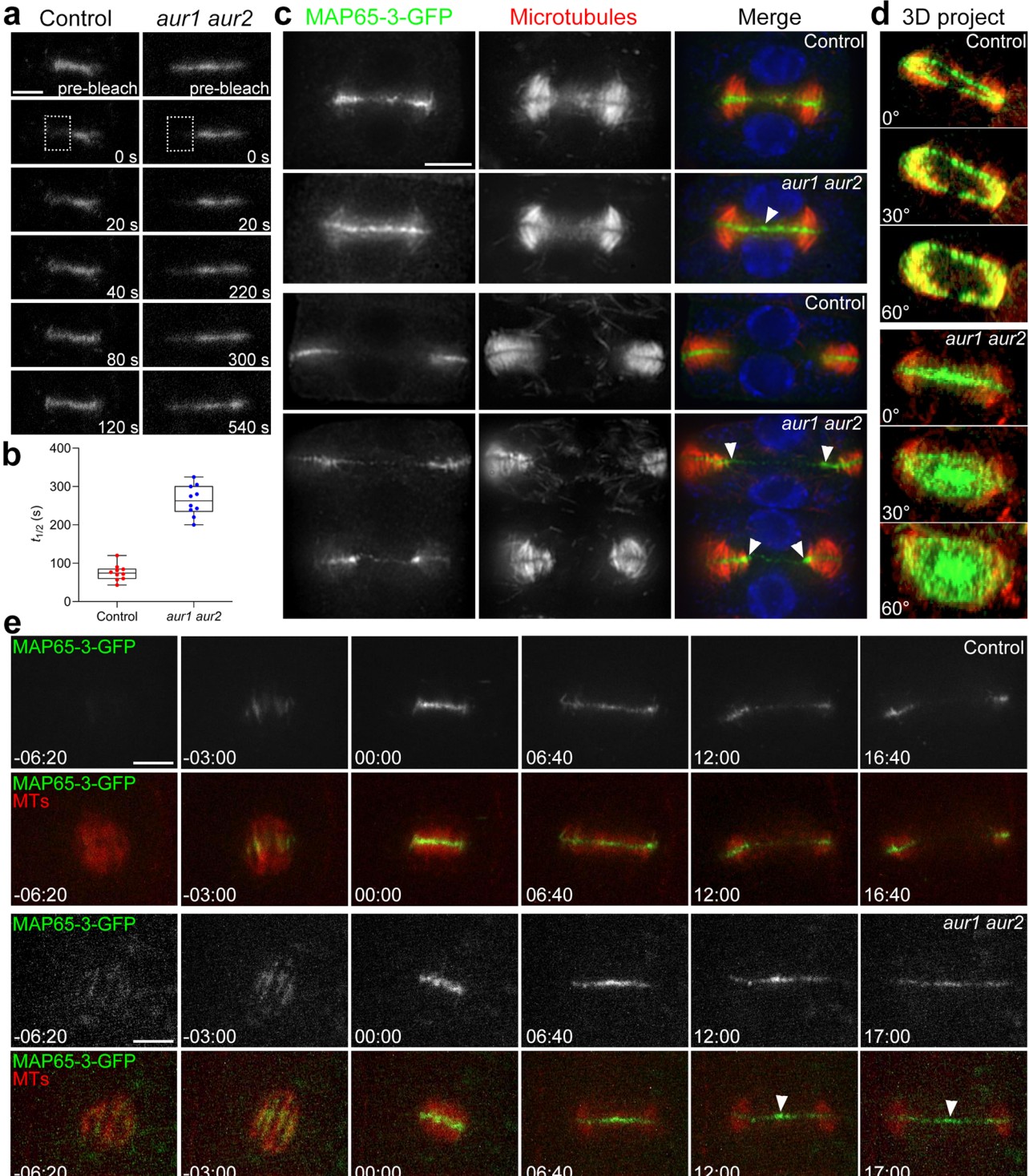

**Fig. 4 | Impaired MAP65-3 turnovers at the phragmoplast midzone in the *aur1 aur2* mutant. a** FRAP analysis of MAP65-3-GFP in *map65-3* (control) and *aur1 aur2* backgrounds. Bleach regions are indicated by a dashed square in first postbleach images. Representative images are acquired from Supplementary Movie 5-6. **b** Quantification of the half-life ($t_{1/2}$) of the fluorescence recovery after bleaching in control and *aur1 aur2* cells ($n = 10$ cells). Data are presented as box-and-whisker plots with individual points, showing the interquartile range (box), the median (horizontal line), and minimum and maximum values (whiskers). **c** Comparison of MAP65-3 distribution in control and *aur1 aur2* plants. Ectopic retentions of MAP65-3 at phragmoplast interior areas are indicated by arrowheads in *aur1 aur2* cells. **d** Rotational views of 3D reconstructed MAP65-3-GFP distribution patterns in control and *aur1 aur2* cells, the rotational angle is denoted. Representative images are acquired from Supplementary Movie 7-8. **e** Live-cell imaging of control and *aur1 aur2* cells expressing MAP65-3-GFP and mCherry-TUB6, representative snapshot images are acquired from Supplementary Movie 9-10. Bars, 5 μm.

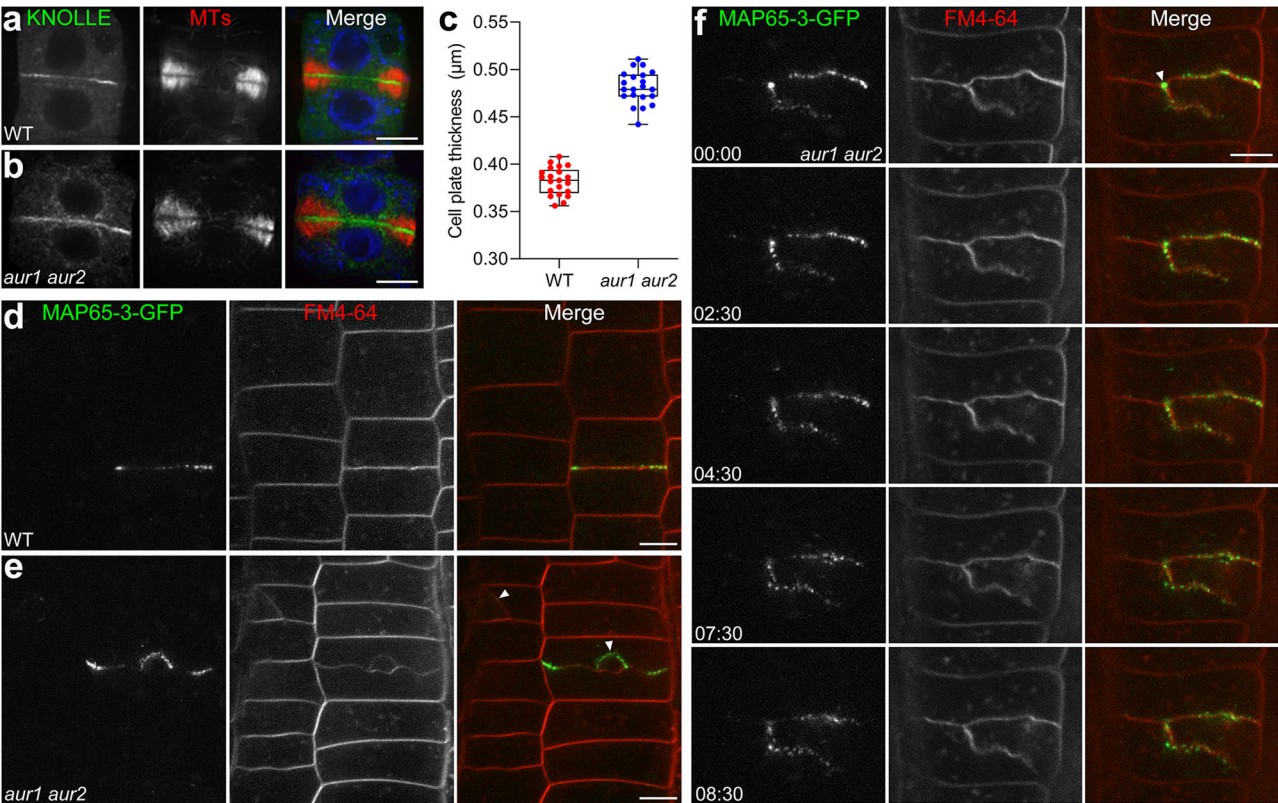

**Fig. 5 | Persistence of MAP65-3 at mature cell plates in *aur1 aur2* cells.**
**a**, **b** Comparative accumulation of KNOLLE in WT (**a**) and *aur1 aur2* (**b**) backgrounds. The merged images have KNOLLE pseudocolored in green, microtubules in red, and DNA in blue. **c** Quantification of the KNOLLE/cell plate thickness in WT and *aur1 aur2* cells (*n* = 21 cells), data are presented as box-and-whisker plots with individual points, showing the interquartile range (box), the median (horizontal line), and minimum and maximum values (whiskers). **d**, **e** MAP65-3-GFP is co-stained with FM4-64 in root tips of 5-day-old WT (**d**) and *aur1 aur2* (**e**) seedlings, branched cell plates in *aur1 aur2* cells are indicated by white arrowheads. **f** Time-lapse images of MAP65-3-GFP at the cell plate in roots of *aur1 aur2* seedlings co-stained with FM4-64, persistent MAP65-3 accumulation at cell plate branching sites is indicated by an arrowhead. Representative snapshot images are acquired from Supplementary Movie 11. Bars, 5 μm.

exhibits limited sequence homology across the Arabidopsis MAP65 family and harbors phosphorylation sites that are known to control MAP65s microtubule bundling activity[11,44]. It has been reported that the C2 region is critical for MAP65-3 to interdigitate antiparallel microtubules in phragmoplasts[42,44]. Grafting the C2 region enables MAP65-1 to acquire an overlapping function with MAP65-3 at the phragmoplast midzone[42]. We find that one of the detected α-Aurora phosphorylation sites, Ser-528, is conserved to a previously reported Ser-532 residue in MAP65-1. The other one, Ser-570, is unique to MAP65-3 versus other MAP65 isoforms. The divergence of phosphorylation sites recognized by the same kinase may distinguish MAP65-3 function or localization from other MAP65 isotypes in plants.

During phragmoplast expansion, the reorganization of microtubules must be tightly coordinated with membrane remodeling to facilitate cell plate assembly. MAP65 isoforms provide a direct physical link between these processes through microtubule bundling and interactions with TRAPPII proteins involved in membrane trafficking at the division site[45]. In the α-Aurora defective plants, a few cells form branched cell plates, correlating with the persistent presence of MAP65-3 at branch site junctions. Failed release of MAP65-3 from microtubules or membranes in *aur1 aur2* plants could potentially induce cell plate branching. Beyond affecting MAP65-3, α-Aurora may directly phosphorylate and regulate proteins integral to membrane dynamics during cytokinesis. α-Aurora may also modulate other MAPs that coordinate microtubule-membrane interactions or link the cytoplasm and cell cortex. Defective regulation of proteins controlling membrane trafficking/remodeling provides an alternative explanation for abnormal cell plate shapes in the α-Aurora mutant. Recently, loss of the phosphoinositide phosphatase *SAC9* was found to cause aberrant enrichment of MAP65-3 at the inner edge of mature cell plates and formation of branched cell plates[46]. Given the similar branched cell plate phenotype in both mutants, exploring the interplay between α-Aurora signaling and phosphoinositide pathways presents an opportunity to delineate their coordinated functions in precisely controlling cell plate expansion and maturation.

In conclusion, we have revealed an aspect of the Arabidopsis α-Aurora kinase in regulating MAP65-3 function, specifically at the phragmoplast midzone. Our findings demonstrate that α-Aurora promotes MAP65-3 turnover through phosphorylation-dependent modulation of its microtubule association ability. While our work has provided insights into MAP65-3 regulation during cytokinesis, we recognize that many additional non-MAP65 proteins make important contributions to phragmoplast structure and function. An outstanding question is whether α-Aurora also targets other midzone-localized proteins involved in phragmoplast organization, such as kinesin motors and microtubule plus-end binding proteins[47–53]. Elucidating new α-Aurora substrates would advance our understanding of the complex signaling networks that precisely coordinate plant cell division.

## Methods

### Plant materials and growth conditions
*Arabidopsis thaliana* materials used in this study include the wild-type (Columbia-0) plants, the *aur1 aur2*[29] and *map65-3* (SALK_022166)[43] mutants. All plants were grown in a chamber under a 16-h-light and 8-h-dark cycle at 22 °C. *Agrobacterium tumefaciens* strain GV3101 was used

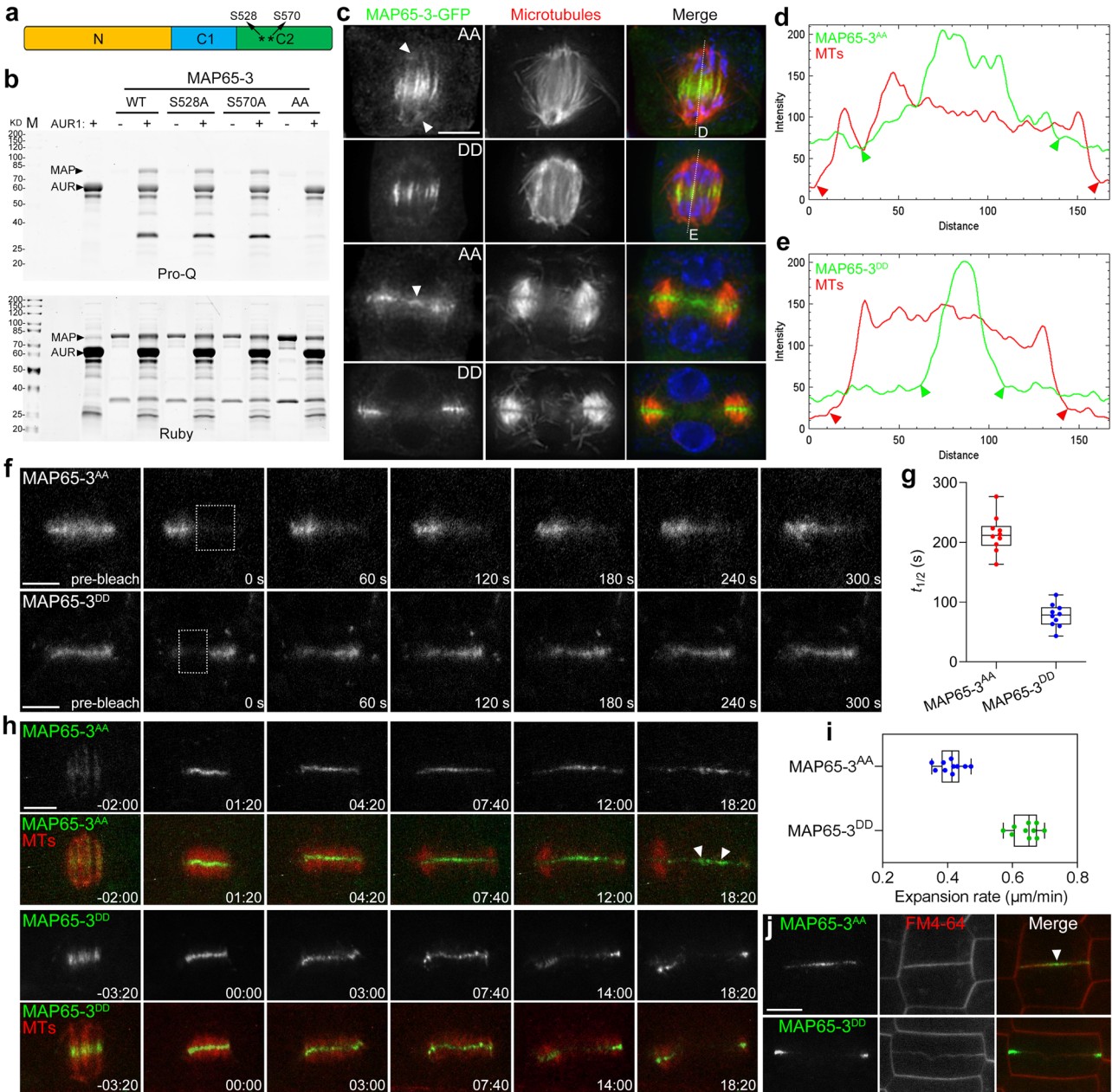

**Fig. 6 | α-Aurora phosphorylates MAP65-3 and affects MAP65-3 associating with microtubules at the phragmoplast midzone. a** Schematic representation of predicated MAP65-3 protein structure. * indicates α-Aurora phosphorylation sites screened by mass spectrometry. **b** In vitro phosphorylation analysis of recombinant MAP65-3 and its site mutation isoforms with or without GST-AUR1. Top gel displays phosphorylation signals by Pro-Q phosphoprotein assay, bottom picture reveals total proteins of the same gel by SYPRO Ruby staining. The experiment was repeated three times with similar results. **c** Distribution of phospho-defective and phospho-mimicking isoforms of MAP65-3 in *map65-3* plants. Arrowheads indicate MAP65-3$^{AA}$ decorating a wider region at anaphase spindles and retaining at phragmoplast interiors. The merged images have MAP65-3 pseudocolored in green, microtubules in red, and DNA in blue. Micrographs are representative of more than 100 cells from three independent lines with similar results.
**d, e** Distribution analysis of MAP65-3$^{AA}$ and MAP65-3$^{DD}$ by fluorescence intensity scans of the regions indicated by dotted lines from merged images. The lowest MAP65-3 phospho-mutant signals are indicated by green arrowheads, and the

edges of central spindle microtubules are indicated by red arrowheads. **f** FRAP analysis of MAP65-3$^{AA}$-GFP and MAP65-3$^{DD}$-GFP in *map65-3* plants. Bleach regions are indicated by a dashed square in the first post-bleach images. Representative images are acquired from Supplementary Movie 12-13. **g** Quantification of the fluorescence recovery half-life ($t_{1/2}$) of MAP65-3$^{AA}$-GFP and MAP65-3$^{DD}$-GFP ($n = 10$ cells), data are presented as box-and-whisker plots with individual points, showing the interquartile range (box), the median (horizontal line), and minimum and maximum values (whiskers). **h** Time-lapse images of MAP65-3$^{AA}$-GFP and MAP65-3$^{DD}$-GFP co-expressing mCherry-TUB6, representative snapshot images are acquired from Supplementary Movie 14-15. **i** Quantitative assessment of the phragmoplast expansion velocity in MAP65-3$^{AA}$ and MAP65-3$^{DD}$ lines ($n = 10$ cells), data are presented as box-and-whisker plots with individual points, showing the interquartile range (box), the median (horizontal line), and minimum and maximum values (whiskers). **j** MAP65-3 phosphovariants co-stained with FM4-64 in the *map65-3* background. The arrowhead indicates MAP65-3$^{AA}$ persisted substantially during cell plate maturation. Bars, 5 μm.

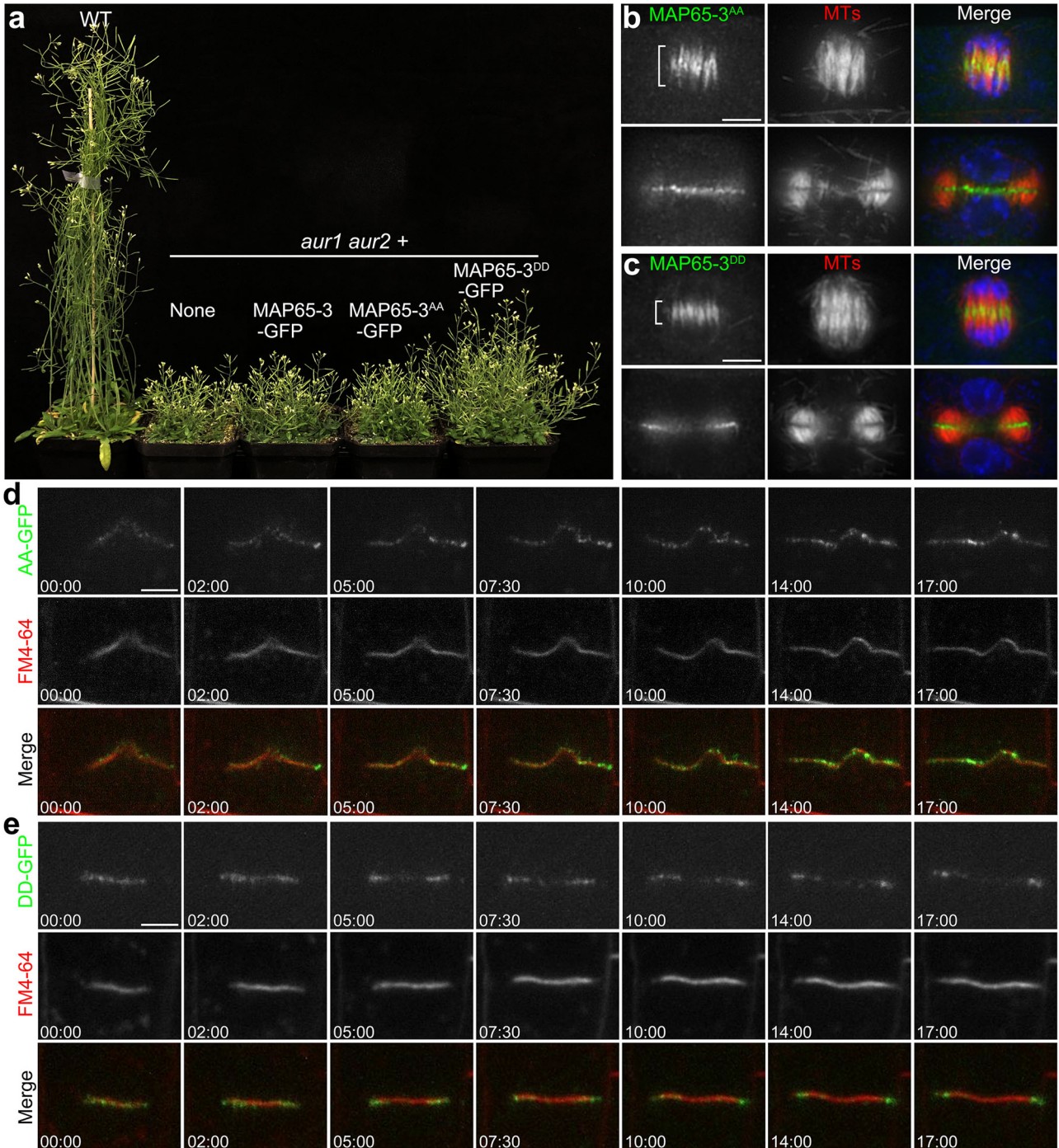

**Fig. 7 | Phospho-mimetic MAP65-3<sup>DD</sup> in *aur1 aur2* plants resembled the behavior of unmodified MAP65-3 in control plants. a** Growth phenotype of 7-week-old plants of WT, *aur1 aur2,* and the mutant expressing different MAP65-3 mutation isoforms. **b, c** Distributing comparison of MAP65-3<sup>AA</sup> (**b**) and MAP65-3<sup>DD</sup> (**c**) in *aur1 aur2* cells. The merged images have MAP65-3<sup>AA</sup> or MAP65-3<sup>DD</sup> pseudocolored in green, microtubules in red, and DNA in blue. Micrographs are representative of more than 100 cells from three independent lines with similar results. **d, e** Time-lapse images of MAP65-3<sup>AA</sup>-GFP (**d**) and MAP65-3<sup>DD</sup>-GFP (**e**) at the cell plate of *aur1 aur2* seedlings co-stained with FM4-64, representative snapshot images are acquired from Supplementary Movie 16-17. Bars, 5 μm.

for floral dipping-based transformation in *A. thaliana*. Seedlings for live-cell imaging and immunolocalization experiments were germinated on a solid medium supplied with 1/2 Murashige Skoog (MS) salt mixture supplied with 0.8% phytagel.

**Plasmid construction**

The genomic fragment of each gene, which contains promoter and coding regions, was amplified and cloned into pDONR221 via BP reaction. The phosphor-mutants of MAP65-3, including single and double mutations, were created via site-directed mutagenesis using the entry vector containing genomic *MAP65-3* as a template. All the resulting entry clones were delivered into pGWB4[54] to get a GFP fusion through LR recombination reactions. The binary vectors containing GFP-AUR1[27], BUB3.1-GFP[37] and GFP/mCherry-TUB6[55] under the control of their native promoters were described as previously. All primers used for plasmid construction are listed in Supplementary Table 1.

## Yeast two-hybrid assay

For the yeast two-hybrid assay, all of the cDNAs tested were amplified and cloned into pDONR221. The entry clones were recombined into the pGADT7-GW (AD) or pGBKT7-GW (BD)[56] through LR recombination. The resulting constructs were transformed into the yeast strain AH109 and were spotted on SD plates without Leu and Trp (-L/-W; control media) or without Leu, Trp, His, and Ade (-L/-W/-H/-A; selection media) and photographed after incubation at 30 °C for 2 days.

## In vitro protein expression and pull-down assay

For recombinant protein production, the coding sequence of AUR1 was cloned into pGEX-KG at the EcoRI & SalI sites. Full-length CDS of MAP65-3 and its various truncations were cloned into pET28a through the BamHI & SalI sites. The recombinant plasmids rendered the expression of GST or His tagged proteins in bacteria host BL21. The fusion proteins were purified using Glutathione Agarose Resins (Yeasen, catalog 20507ES) or Ni-NTA Agarose Resins (Yeasen, catalog 20502ES).

For pull-down assay, purified His-fused MAP65-3 or its truncations were incubated with equal amounts of GST-AUR1 beads in GST pull-down binding buffer (20 mM Tris-HCl pH 8.0, 150 mM NaCl, 0.2% Triton) at 4 °C for 2 h. After washing 5 times with washing buffer (20 mM Tris-HCl pH 8.0, 150 mM NaCl, 0.1% Triton), the beads were collected, boiled in 50 μL 1× SDS loading buffer for 10 min at 100 °C, and then examined by immunoblot using anti-His and anti-GST antibodies (Yeasen, catalog 30401ES and 30901ES, 1:10000).

## Immunolocalization and fluorescence microscopy

For immunofluorescence staining experiments, root meristematic cells from 5-day-old seedlings were excised and fixed for 45 min in PME (50 mM PIPES, pH 6.9, 5 mM MgSO4, 1 mM EGTA, and 4% formaldehyde) at room temperature, following a treatment of 1% Cellulase (Onozuka RS, Yakult) for 10 min. The fixed cells were then immobilized on slides and treated with 0.5% Triton X-100 for 15 min and methanol for 10 min prior to incubation with antibodies. Primary antibodies used in this study included GFP recombinant rabbit monoclonal antibody (Thermo Fisher, catalog G10362, 1:400), KNOLLE rabbit polyclonal antibody (PhytoAB, catalog PHY2910A, 1:400), and DM1A mouse α-tubulin monoclonal antibody (Abcam, catalog ab7291, 1:1000). Secondary antibodies were Alexa fluor 488-conjugated goat anti-rabbit IgG and Alexa fluor 555-conjugated goat anti-mouse IgG (Thermo Fisher, catalog A32731 and A32727, 1:1000). Stained cells were observed under an Eclipse 600 microscope equipped with 100x Plan-Apo objective (NA 1.45, Nikon). Images were acquired by a panda sCMOS camera (PCO Imaging).

For live-cell observation, root meristematic cells of 5-day-old seedlings were observed using a LSM880 spinning-disk confocal microscope equipped with a 100x oil-immersion objective (NA 1.4, Zeiss). GFP was excited with a 488 nm laser and detected between 500-550 nm. mCherry and FM4-64 were excited with a 561 nm laser and detected between 570–650 nm. Time-lapse images were acquired using the ZEN software package (Zeiss) and processed in ImageJ.

For FM4-64 staining, 5-day-old seedlings were incubated in 2 μM FM4-64 (Thermo Fisher, catalog T13320) for 10 min and then briefly washed with half-strength liquid MS medium. The seedlings were then scanned with the LSM880 confocal microscope every 30 seconds until the indicated time point. To determine phragmoplast/cell plate expansion rates, both leading edges of phragmoplast microtubules or FM4-64 signal are tracked over time in different cells.

## Transient expression of fusion proteins in tobacco leaves

For transient expression assay, wild-type *Nicotiana benthamiana* plants were grown in soil at 26 °C under a 16-h light and 8-h dark cycle. Leaves of 5-week-old plants were infiltrated with *A. tumefaciens* strain GV3101 carrying the plasmids of interest. *A. tumefaciens* cultures were grown overnight at 28 °C, pelleted, washed twice with 1/2 liquid MS medium, and resuspended to OD600 0.8 before infiltration.

BiFC assay was carried out using the N- or C-terminus of YFP as described previously[56]. Briefly, the entry clones containing coding sequences of AUR1, AUR3, and MAP65-3 were recombined with pGTQL1211YN or pGTQL1221YC via LR clonase. *A. tumefaciens* strains carrying nYFP or cYFP constructs were mixed and infiltrated into the young leaves of 5-week-old *N. benthamiana* plants. After 48 hours, YFP signals were excited with a 488 nm laser line, whereas the detected emission wavelength range was 500–550 nm.

## Photobleaching experiments

FRAP experiments for the MAP65-3-GFP were conducted on a LSM880 laser scanning confocal microscope equipped with a 40x water immersion objective (NA 1.2, Zeiss). A certain region of interest was bleached using a 488-nm laser line for 100 iterations with 100% laser power. Fluorescent signal recovery was imaged every 10 s and the average fluorescence intensity of the bleached region was analyzed in the ZEN software.

## In vitro kinase assay and identification of phosphorylation sites

For in vitro kinase assays, 0.5 μg His-MAP65-3 and its site mutation isoforms were incubated with or without 2 μg GST-AUR1 in 40 μL of kinase buffer [10 mM HEPES at pH 7.5, 20 mM MgCl$_2$, 1 mM DTT, 5 mM EGTA and 0.1 mM ATP]. After incubation at 37 °C for 60 min, the reactions were stopped by adding 10 μL of 5×SDS loading buffer and boiling for 10 min. Samples were resolved by SDS-PAGE, and phosphorylation was detected by the Pro-Q™ Diamond Phosphoprotein Gel Staining Kit (Thermo Fisher, catalog P33300).

To identify phosphorylation sites, the gels were stained with PageBlue™ protein staining solution (Thermo Fisher, catalog 24620), the stained bands were then excised from the gels and subjected to in-gel trypsin digestion. This involved incubating the gel pieces with sequencing-grade modified trypsin at 37 °C for 16 hours to enzymatically break down the proteins into peptides. The resulting peptide mixture contained phosphorylated peptides which were analyzed by liquid chromatography tandem mass spectrometry by Shanghai Luming Biological Technology Co., Ltd.

## Reporting summary

Further information on research design is available in the Nature Portfolio Reporting Summary linked to this article.

## Data availability

All relevant data are available within the manuscript and its supplementary materials. Source data are provided in this paper.

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

## Acknowledgements

We thank Dr. Daniël Van Damme (Ghent University) and Dr. Tsuyoshi Nakagawa (Shimane University) for sharing the *aur1 aur2* mutant and pGWB vectors. This study was supported by the National Natural Science Foundation of China (U22A20494, 32270354, and 31900163), the Natural Science Foundation of Sichuan Province (2022NSFSC1651), Institutional Research Fund of Sichuan University (2020SCUNL212), and Sichuan Forage Innovation Team Program (sccxtd-2020-16).

## Author contributions

X.D., B.L. and H.L. conceived and supervised the project, X.D. and Y.X. performed most of the experiments, X.D., Y.X. and X.T. analyzed the data, and X.D., B.L. and H.L. wrote the manuscript. All authors have read and approved the paper submission.

## Competing interests

The authors declare no competing interests.
