## [Peer Review File · Nature Communications]

Arabidopsis α -Aurora Kinase Plays a Role in Cytokinesis through Regulating MAP65-3 Association with Microtubules at Phragmoplast MidzoneReviewer #1 (Remarks to the Author):

The phragmoplast, a plant-specific microtubule-based structure, is constructed as a scaffold for the formation of the cell plate during cytokinesis. Throughout cytokinesis the cell plate materials are transported and fused. Although the mitogen-activated protein kinase (MAPK) cascade is a well-known regulatory mechanism of phragmoplast expansion, the involvement of other kinase(s) in MAP65-mediated phragmoplast regulation has been speculated. In this study, the authors focused on Arabidopsis α -Aurora kinase, AUR1, which is localized at the spindle poles during mitosis and relocated to the phragmoplast midzone during cytokinesis. The interaction between AUR1 and MAP65-3 was confirmed by yeast two-hybrid, BiFC, and in vitro pull-down assays. The authors found that MAP65-3 localizes to the phragmoplast midzone in the control, while it exhibits a broader localization pattern in the phragmoplast in the hypomorphic α -Aurora kinase mutant. In the *aur1 aur2* mutant, the phragmoplast expansion and MAP65-3 turnover rates at the phragmoplast midzone decreased, and the antiparallel microtubules at the leading edge of the expanding phragmoplast were disengaged. Furthermore, the authors revealed that the microtubule-binding domain of MAP65-3 is phosphorylated by AUR1 in vitro. Complementation analysis using phospho-mimicking MAP65-3DD and non-phosphorylatable MAP65-3AA mutant proteins showed that the phosphorylation of the microtubule-binding domain is important for MAP65-3's midzone localization and rapid turnover in the phragmoplast. This study elucidated an α -Aurora kinase-mediated MAP65-3 regulation, where phosphorylation on MAP65-3 defines its localization and ensures rapid phragmoplast expansion during cytokinesis.

Overall assessment

The key findings presented in this study, if validated by further experimentation, as described below, are novel and provide new insights into the molecular mechanisms underlying phragmoplast formation in plants. However, a conceptually similar study was published several years ago by the Van Damme laboratory in Plant Physiology, wherein α -Aurora phosphorylation of MAP65-1 was shown critical for its localization and function during cell division (Boruc et al., 2017, Plant Physiol). Although the mitotic functions of MAP65-1 and MAP65-3 are slightly different, conceptual advances made in the present study are limited.

Specific points

1. The methods must be described in more detail. For example, the buffers used for protein purification and the protein concentrations in the in vitro assays have not been described.
2. FRAP experiments: Is the turnover rate of MAP65-3 consistent throughout phragmoplast expansion? Otherwise, the data could be interpreted differently. The turnover rate should be examined during the early, middle, and late phases of cytokinesis. Alternatively, the turnover rate should be measured and compared at a consistent time point during phragmoplast expansion (e.g., 5 min after anaphase onset).
3. The purity of the recombinant proteins used in the in vitro phosphorylation assay was extremely low (Fig. 6B). Further protein purification methods, such as gel filtration chromatography, should be used.
4. The authors describe the cytokinesis duration as follows: "In wild-type cells, cytokinesis (from the stage of phragmoplast initiation to completion of phragmoplast disassembly) lasted approximately 20 min ...a significantly longer time of 35 min during cytokinesis". This is important information for data interpretations and needs to be presented in a more quantitative manner (mean \pm SD, sample size [n]).
5. The authors used three different terms to describe the phragmoplast middle plane, where cell plate assembly occurs (midzone, midline, and equatorial zone). These terms should be unified to avoid confusions unless they are intended to describe different structures.
6. Arabidopsis BUB3 has been shown to regulate MAP65-3 localization (Zhang et al., 2018). Since the MAPK cascade also regulates MAP65 activity, multiple mechanisms are involved in MAP65 regulation. The manuscript should discuss how the mechanism revealed in this study is distinct from or similar to other mechanisms.

7. It is not very clear which phragmoplast morphology the authors referred to as "early phragmoplast," "late phragmoplast," "ring-shaped," and "a solid disk." They should be clearly indicated in an early figure (e.g., Fig. 2).

8. Scale bars are missing in e.g., Figs. 2C, 5A, 6F, 7A, S1B, and S2. In addition, molecular size information should be given next to the protein gel.

9. The distinction between the cytokinetic function of MAP65-3 and other MAP65s is explained by citing REF 41–43 in the discussion section. This explanation might be moved to the Introduction, which clarifies why the authors focused on MAP65-3 rather than on any other MAP65s.

10. How were the phragmoplast expansion rates quantified? Were both leading edges of a phragmoplast measured in a cell? Details on the analyses need to be clarified in the methodology section.

11. Fig. S1A, BD-AUR3: Are these panels placed in the correct order?

12. Many typos and grammatical errors have been identified. Gross editing of the manuscript is necessary.

Reviewer #2 (Remarks to the Author):

This work explores interaction between microtubule protein MAP65-3 and α -Aurora kinase during cytokinesis. The role of α -Aurora in cytokinesis and in phosphorylation of a close ortholog of MAP65-1 was shown before (Boruc et al., 2017). Here another member of the MAP65 gene family, MAP65-3 was shown to be a substrate for α -Aurora kinase. In addition, the authors demonstrate direct interaction between MAP65-3 and α -Aurora kinase. Despite identification of a new substrate of α -Aurora kinase is significant, it is necessary to determine the functional significance of the phosphorylation of MAP65-3 and the interaction between MAP65-3 and α -Aurora kinase by more detailed analysis of the mutant and transgenic lines. Below I provide specific suggestions.

The first major conceptual shortfall is the assumption that α -Aurora has only one substrate. This assumption has not been experimentally proven and contradicts the previous finding that α -Aurora phosphorylates MAP65-1. The authors assume that cytokinetic defects are caused by the lack of phosphorylation of two residues on MAP65-3. It is highly likely that α -Aurora phosphorylates other proteins and disruption of these events contributes to the cell division defects in *aur1aur2*. The Authors could and should compare phosphorylation events in control and *aur1aur2* using phosphoproteomics.

Second significant conceptual weakness is the assumption that MAP65-3 only interacts with α -Aurora. It has been shown that MAP65-3 interacts with POK2 and TRAPP1 (Herrmann et al., 2018; Steiner et al., 2016). Both of these proteins are essential for cytokinesis. Thus, defects in MAP65-3 localization or phosphorylation could affect interaction with these proteins. If MAP65-3 was the only substrate of α -Aurora, then the phosphomimetic version (MAP65-3DD) should be unable to rescue the *map65-3* phenotype. However, Supplemental Figure 3 shows that the DD mutant rescues *map65-3* root growth phenotype. It means that phosphorylation of MAP65-3 by *aurora* kinase may not be responsible for cytokinetic defects in *aur1aur2*. This outcome is more consistent with the hypothesis that phosphorylation of MAP65-3 interferes with the cell plate assembly through TRAPP1, POK2, or other components of the cell plate assembly. The authors should examine cell plate morphology in *aur1aur2*, AA, and DD mutants. They also should examine localization of POK1 and TRS120 in the mutant background.

Another fact that questions significance of MAP65-3 phosphorylation by α -Aurora is differences between phenotype of *aur1aur2* mutant and line expressing MAP65-3AA. In *aur1aur3* MAP65-3 does not localize to the phragmoplast tip (Figure 5C) whereas MAP65-3AA labels the entire phragmoplast (Figure 6C). Considering this discrepancy the authors should explore other substrates of α -Aurora kinase by comparing the phosphoproteome of Col-0 and *aur1aur2*.

Inconsistency of experimental approaches to protein localization is a significant concern. Many localization studies are performed using chemical fixation, which is prone to artefacts. As a consequence there are several confusing results. For example, the lack of MAP65-3 localization at the phragmoplast tip in Figure 5C is a common artefact of chemical fixation. The authors should generate *aur1aur2* mutants expressing MAP65-3 and tubulin markers and analysed co-localization of MAP65-3 for these experiments. It is also important to analyse co-localization of cell plate assembly markers KNOLLE and MAP65-3 on Col-0 and *aur1aur2* backgrounds. Another example is Figure 6. The top row in panel F is labelled as the MAP65-3AA mutant whereas the localization pattern is identical to the MAP65-3DD mutant in Figure 6C. The bottom row in panel F is labelled MAP65-3DD, but the localization pattern is identical to the AA mutant in panel C. These comparisons should be performed in living cells during phragmoplast expansion.

Expansion rate in Figure 5 was measured incorrectly. As Figure 5C shows that MAP65-3 may not always follow the phragmoplast tip, the expansion rate should be measured using a microtubule marker.

The authors should demonstrate how phosphorylation affects activity of MAP65-3. Slower turnover in the midzone in Figure 6G could be a consequence of defective cell plate assembly or association with other proteins. These possibilities should be verified. As authors already have lines expressing GFP-fusions of mutants, a simple-immunoprecipitation will provide information about interactomes of the mutants.

Analysis of interaction between MAP65-3 and α -Aurora is preliminary and requires functional characterization. The authors should examine the function of interaction between α -Aurora and MAP65-3 by complementing *aur1aur2* with α -Aurora mutant that does not bind MAP65-3 and complement *map65-3* with MAP65-3 mutant that does not bind Aurora A.

Association of MAP65-3 with microtubules in *aur1aur2* suggests that MAP65-3 can bind parallel microtubules. However, during cytokinesis in *aur1aur2* MAP65-3 binds only antiparallel microtubules. These data suggest that phosphorylation by α -Aurora changes microtubule bundling mechanisms by MAP65-3. This should be tested by determining polarity of microtubule bundles produced by the mutants and wild type MAP65-3 in vitro.

Minor weaknesses:

Figure 1 has been published before and should be removed from the paper.

References

- Boruc, J., A.K. Weimer, V. Stoppin-Mellet, E. Mylle, K. Kosetsu, C. Cedeno, M. Jaquinod, M. Njo, L. De Milde, P. Tompa, N. Gonzalez, D. Inze, T. Beeckman, M. Vantard, and D. Van Damme. 2017. Phosphorylation of MAP65-1 by Arabidopsis Aurora Kinases Is Required for Efficient Cell Cycle Progression. *Plant Physiol.* 173:582-599.
- Herrmann, A., P. Livanos, E. Lipka, A. Gadeyne, M.T. Hauser, D. Van Damme, and S. Muller. 2018. Dual localized kinesin-12 POK2 plays multiple roles during cell division and interacts with MAP65-3. *Embo Reports.* 19.
- Steiner, A., K. Rybak, M. Altmann, H.E. McFarlane, S. Klaeger, N. Nguyen, E. Facher, A. Ivakov, G. Wanner, B. Kuster, S. Persson, P. Falter-Braun, M.T. Hauser, and F.F. Assaad. 2016. Cell cycle-regulated PLEIADE/AtMAP65-3 links membrane and microtubule dynamics during plant cytokinesis. *Plant J:*531-541.

Reviewer #1 (Remarks to the Author):

The phragmoplast, a plant-specific microtubule-based structure, is constructed as a scaffold for the formation of the cell plate during cytokinesis. Throughout cytokinesis the cell plate materials are transported and fused. Although the mitogen-activated protein kinase (MAPK) cascade is a well-known regulatory mechanism of phragmoplast expansion, the involvement of other kinase(s) in MAP65-mediated phragmoplast regulation has been speculated. In this study, the authors focused on Arabidopsis α -Aurora kinase, AUR1, which is localized at the spindle poles during mitosis and relocated to the phragmoplast midzone during cytokinesis. The interaction between AUR1 and MAP65-3 was confirmed by yeast two-hybrid, BiFC, and in vitro pull-down assays. The authors found that MAP65-3 localizes to the phragmoplast midzone in the control, while it exhibits a broader localization pattern in the phragmoplast in the hypomorphic α -Aurora kinase mutant. In the aur1 aur2 mutant, the phragmoplast expansion and MAP65-3 turnover rates at the phragmoplast midzone decreased, and the antiparallel microtubules at the leading edge of the expanding phragmoplast were disengaged. Furthermore, the authors revealed that the microtubule-binding domain of MAP65-3 is phosphorylated by AUR1 in vitro. Complementation analysis using phospho-mimicking MAP65-3DD and non-phosphorylatable MAP65-3AA mutant proteins showed that the phosphorylation of the microtubule-binding domain is important for MAP65-3's midzone localization and rapid turnover in the phragmoplast. This study elucidated an α -Aurora kinase-mediated MAP65-3 regulation, where phosphorylation on MAP65-3 defines its localization and ensures rapid phragmoplast expansion during cytokinesis.

Overall assessment

The key findings presented in this study, if validated by further experimentation, as described below, are novel and provide new insights into the molecular mechanisms underlying phragmoplast formation in plants. However, a conceptually similar study was published several years ago by the Van Damme laboratory in Plant Physiology, wherein α -Aurora phosphorylation of MAP65-1 was shown critical for its localization and function during cell division (Boruc et al., 2017, Plant Physiol). Although the mitotic functions of MAP65-1 and MAP65-3 are slightly different, conceptual advances made in the present study are limited.

Response: We appreciate the reviewer raising this concern regarding conceptual overlap with previous findings linking α -Aurora and MAP65-1, and agree that additional experimentations would strengthen the study. Meanwhile, we hold the opinion that our identification of MAP65-3 as a novel substrate and key effector of α -Aurora at the phragmoplast midzone represents a significant conceptual advance that provides new insights into phragmoplast organization. While α -Aurora has been previously shown to phosphorylate MAP65-1, MAP65-1 exhibits distinct localization from α -Aurora during cytokinesis. MAP65-1 localizes throughout the phragmoplast, whereas α -Aurora specifically accumulates at the phragmoplast midzone. Thus, we propose α -Aurora exerts its effects on cytokinesis rely on its phosphorylation of MAP65-3 instead of MAP65-1 because MAP65-3 shows exclusive localization at or near microtubule plus ends in the phragmoplast. By uncovering a new pathway involving MAP65-3 as a key intermediary, we provide novel understanding of how α -Aurora signaling is translated into

accelerated phragmoplast expansion and progression through cytokinesis. This significantly expands knowledge of α -Aurora's diverse functions during cell division in plants.

Specific points

1. *The methods must be described in more detail. For example, the buffers used for protein purification and the protein concentrations in the in vitro assays have not been described.*

Response: We appreciate the critique that providing more in-depth methods is necessary for our work. We have thoroughly addressed this concern by adding additional details on “Methods” in the revised manuscript.

2. *FRAP experiments: Is the turnover rate of MAP65-3 consistent throughout phragmoplast expansion? Otherwise, the data could be interpreted differently. The turnover rate should be examined during the early, middle, and late phases of cytokinesis. Alternatively, the turnover rate should be measured and compared at a consistent time point during phragmoplast expansion (e.g., 5 min after anaphase onset).*

Response: We appreciate the reviewer's suggestion to examine MAP65-3 turnover at multiple time points during phragmoplast expansion and performed further experiments to confirm the consistency of MAP65-3 turnover rate. As shown in Supplemental Figure S7, the turnover rate of MAP65-3 is also reduced in *aur1 aur2* plants at late stages of cytokinesis when compared with control plants. Quantitatively, in both control and *aur1 aur2* cells, the half-life of fluorescence recovery after photobleaching (FRAP) at different cytokinesis stages shows little difference. Previously we chose to focus on early cytokinesis stages because at this time MAP65-3 retains a substantial disk-like signal that is straightforward to track. At late phases of cytokinesis, sometimes it can be difficult to distinguish whether regions lacking signal are caused by photobleaching or expansion of the MAP65-3 ring-edge.

3. *The purity of the recombinant proteins used in the in vitro phosphorylation assay was extremely low (Fig. 6B). Further protein purification methods, such as gel filtration chromatography, should be used.*

Response: To address this point, we have optimized our purification methods and achieved highly pure protein preparations, especially for AUR1 (Figure 6B). The improved reagents in kinase assays resulted in verification of our original results, supporting that MAP65-3 is a substrate of Aurora kinases.

4. *The authors describe the cytokinesis duration as follows: “In wild-type cells, cytokinesis (from the stage of phragmoplast initiation to completion of phragmoplast disassembly) lasted approximately 20 min ...a significantly longer time of 35 min during cytokinesis”. This is important information for data interpretations and needs to be presented in a more quantitative manner (mean \pm SD, sample size [n]).*

Response: The duration time described here presents the cells with similar sizes shown in Figure 1A-B. In actuality, cells of different sizes undergo cytokinesis over vastly different timescales. For instance, flat cells with broader cell plate formation area always exhibit longer cytokinesis durations than narrow cells. Therefore, we prefer to quantify phragmoplast expansion rates, which are scarcely influenced by

cell size, to depict cytokinesis progression. We appreciate the reviewer highlighting the need to provide quantitative details. In our revised manuscript, we have included mean data with standard deviations and sample sizes for the expansion rate analyses.

5. *The authors used three different terms to describe the phragmoplast middle plane, where cell plate assembly occurs (midzone, midline, and equatorial zone). These terms should be unified to avoid confusions unless they are intended to describe different structures.*

Response: We appreciate this feedback regarding inconsistent description. We have unified descriptions to “midzone” in the revised manuscript to avoid confusion.

6. *Arabidopsis BUB3 has been shown to regulate MAP65-3 localization (Zhang et al., 2018). Since the MAPK cascade also regulates MAP65 activity, multiple mechanisms are involved in MAP65 regulation. The manuscript should discuss how the mechanism revealed in this study is distinct from or similar to other mechanisms.*

Response: We appreciate the reviewer highlighting the need to discuss our findings in context of other known mechanisms regulating MAP65-3 and phragmoplast microtubule dynamics. MAP65-3 is a common target for regulating phragmoplast microtubule dynamics, and always functions as a microtubule bundling factor required for other proteins such as BUB3s and Kinesin-12s to the phragmoplast midzone. Our work identifies α -Aurora kinase as a direct upstream regulator of MAP65-3. The MAP kinase cascade also regulates MAP65 activity, providing another mode of control that could function alongside Aurora kinase signaling. Thus MAP65-3 emerges as a focal point for orchestrating microtubule organization at the phragmoplast midzone, under influence of several pathways. A sophisticated system of checks and balances, including α -Aurora and MAP kinases, may act cooperatively or redundantly to govern MAP65-3 and ensure proper phragmoplast progression. We provide new mechanistic understanding of how α -Aurora kinase modulates MAP65-3 as a part of this system, advancing knowledge of the regulatory framework directing plant cytokinesis. Our future work is aimed at determining how these different mechanisms are integrated to control phragmoplast organization and MAP65 dynamics for the completion of cell division. These points have been discussed in the revised manuscript.

7. *It is not very clear which phragmoplast morphology the authors referred to as “early phragmoplast,” “late phragmoplast,” “ring-shaped,” and “a solid disk.” They should be clearly indicated in an early figure (e.g., Fig. 2).*

Response: We thank the reviewer pointing out lack of clarity for phragmoplast morphology and we have unified the definition of those stages in Figure 1.

8. *Scale bars are missing in e.g., Figs. 2C, 5A, 6F, 7A, S1B, and S2. In addition, molecular size information should be given next to the protein gel.*

Response: Sorry for the mistakes. In the revised manuscript, scales bars are added in those figures, and a protein marker is also included within protein/kinase assay gels.

9. The distinction between the cytokinetic function of MAP65-3 and other MAP65s is explained by citing REF 41–43 in the discussion section. This explanation might be moved to the Introduction, which clarifies why the authors focused on MAP65-3 rather than on any other MAP65s.

Response: We appreciate the reviewer suggesting clarification of why MAP65-3 is the focus of our study would be helpful for readers, and agree introducing this distinction earlier would provide important context. In our revised manuscript, we have moved the explanation to “Introduction”.

10. How were the phragmoplast expansion rates quantified? Were both leading edges of a phragmoplast measured in a cell? Details on the analyses need to be clarified in the methodology section.

Response: To determine the phragmoplast expansion rates, both leading edges of phragmoplast microtubule arrays are tracked over time in wild-type and mutant cells. Details on the analyses have been added to “Methods”.

11. Fig. S1A, BD-AUR3: Are these panels placed in the correct order?

Response: Sorry for our negligence, the positions of the BD-AUR3 data panels in Fig. S1A were indeed reversed. We have corrected the figure.

12. Many typos and grammatical errors have been identified. Gross editing of the manuscript is necessary.

Response: We have performed a comprehensive review of the manuscript to identify and correct typos, grammatical errors, or awkward/unclear phrasing detected.

Reviewer #2 (Remarks to the Author):

This work explores interaction between microtubule protein MAP65-3 and α -Aurora kinase during cytokinesis. The role of α -Aurora in cytokinesis and in phosphorylation of a close ortholog of MAP65-1 was shown before (Boruc et al., 2017). Here another member of the MAP65 gene family, MAP65-3 was shown to be a substrate for α -Aurora kinase. In addition, the authors demonstrate direct interaction between MAP65-3 and α -Aurora kinase. Despite identification of a new substrate of α -Aurora kinase is significant, it is necessary to determine the functional significance of the phosphorylation of MAP65-3 and the interaction between MAP65-3 and α -Aurora kinase by more detailed analysis of the mutant and transgenic lines. Below I provide specific suggestions.

*The first major conceptual shortfall is the assumption that α -Aurora has only one substrate. This assumption has not been experimentally proven and contradicts the previous finding that α -Aurora phosphorylates MAP65-1. The authors assume that cytokinetic defects are caused by the lack of phosphorylation of two residues on MAP65-3. It is highly likely that α -Aurora phosphorylates other proteins and disruption of these events contributes to the cell division defects in *aur1aur2*. The Authors could and should compare phosphorylation events in control and *aur1aur2* using phosphoproteomics.*

Response: We apologize for any misunderstanding, but we did not intend to leave the impression that α -Aurora has only one substrate during cell division. We agree α -Aurora likely phosphorylates other proteins during cytokinesis. Our goal is to uncover a new pathway involving MAP65-3 as a key intermediary, providing novel understanding of how α -Aurora signaling drives cytokinesis progression in plants. We hope this clarifies that we do not assume α -Aurora has a single substrate, but seek to identify MAP65-3 as a new substrate important for its role at the phragmoplast midzone. We focus on this newly identified pathway, but recognize α -Aurora likely acts through other mechanisms as well.

Phosphoproteomics analysis is a powerful approach to thoroughly understand Aurora functions. Given that α -Aurora kinase expresses only in dividing cells, and *aur1 aur2* is a knockdown mutant because totally knockout of α -Aurora results in lethality in Arabidopsis. Thus it is particularly challenging for us to allow detection of differences in phosphorylation between wild-type and mutant by mass spectrometry. Indeed, identifying and comprehensively exploring α -Aurora's network of substrates and mechanisms of regulation during cell division are planned for our future studies, however, such a massive research obviously would require years of intensive work. Our identification of MAP65-3 as a new α -Aurora substrate that plays an important role in phragmoplast progression, as a focused study, significantly advances understanding of how this spindle pole and phragmoplast midzone dual-localized kinase promotes timely completion of cytokinesis during plant cell proliferation. Future work building upon our findings should help integrate additional components of this network to provide a systems-level view of α -Aurora function in cell division.

Second significant conceptual weakness is the assumption that MAP65-3 only interacts with α -Aurora. It has been shown that MAP65-3 interacts with POK2 and TRAPP11 (Herrmann et al., 2018; Steiner et al., 2016). Both of these proteins are essential for cytokinesis. Thus, defects in MAP65-3 localization or

phosphorylation could affect interaction with these proteins. If MAP65-3 was the only substrate of α -Aurora, then the phosphomimetic version (MAP65-3^{DD}) should be unable to rescue the *map65-3* phenotype. However, Supplemental Figure 3 shows that the DD mutant rescues *map65-3* root growth phenotype. It means that phosphorylation of MAP65-3 by aurora kinase may not be responsible for cytokinetic defects in *aur1aur2*. This outcome is more consistent with the hypothesis that phosphorylation of MAP65-3 interferes with the cell plate assembly through TRAPP^{II}, POK2, or other components of the cell plate assembly. The authors should examine cell plate morphology in *aur1aur2*, AA, and DD mutants. They also should examine localization of POK2 and TRS120 in the mutant background.

Response: We appreciate the thoughtful feedback, which has helped identify some misunderstandings in our initial responses. In our manuscript we did not state MAP65-3 only interact with α -Aurora during cytokinesis. MAP65-3 always functions as a scaffold to recruit other proteins to the phragmoplast midzone. While we show α -Aurora phosphorylates MAP65-3, the phosphorylation may not affect the interaction of MAP65-3 with other proteins, e.g., BUB3.1 and POK2.

Despite multiple attempts, we are unable to propagate full genomic clones for POK2. Therefore, we chose to examine Kinesin-12A, another kinesin-12 family member that also localizes to the phragmoplast midzone in a MAP65-3-dependent manner. We deliver these proteins (BUB3.1 and Kinesin-12A) thought to associate with MAP65-3, as well as TRS120, to wild-type and the *aur1 aur2* mutant plants. Interestingly, we find those proteins still localize to the phragmoplast midzone in *aur1 aur2* cells (Supplemental Figure S5), providing strong evidence that MAP65-3 continues to act upstream in recruiting these proteins in the *aur1 aur2* mutant background.

We propose α -Aurora phosphorylation weakens MAP65-3 association with interdigitated microtubules, accelerating their turnover and phragmoplast expansion. Now we provide new evidence to further support our hypothesis. As shown in Figure 7, expressing the phospho-mimicking MAP65-3^{DD} suppresses the bushy phenotype and slow phragmoplast expansion in the *aur1 aur2* mutant (Figure 7). Furthermore, MAP65-3^{DD} localization in *aur1 aur2* cells is comparable to its native form in control cells (Figure 7). MAP65-3^{DD} rescuing phenomenon demonstrates that mimicking constitutive MAP65-3 phosphorylation partially recapitulates some aspects of α -Aurora-mediated signaling when the α -Aurora activity is compromised. This outcome supports phosphor-regulation of MAP65-3 dynamics as a mechanism by which α -Aurora promotes phragmoplast expansion.

The reviewers' comments also prompt us to examine cell plate morphology in different plants and find most *aur1 aur2* cells are able to form mature but slightly thicker cell plates in comparison with wild-type cells (Figure 5). The expansion rate of the cell plate in *aur1 aur2* cells is also slowed down, similar to the slower phragmoplast expansion rate observed. Furthermore, we detect branched cell plates in some cells in the *aur1 aur2* mutant that are never seen in the wild-type control. Colocalization

experiments demonstrate that the aberrant cell plate always correlates with the persistence of MAP65-3 on branch initiation sites or the center of the maturing cell plate (Figure 5). Persistent MAP65-3 presence at branch sites/cell plate center in *aur1 aur2* cells could potentially induce cell plate branching. Based on the current knowledge, we could not exclude that α -Aurora plays a role during the fusion of the cell plate as its cell plate localization is not dependent on MAP65-3. However, identification of the molecular machinery controlling the cell plate development by α -Aurora requires extensive work beyond the current study. Here, we focus on and provide a compelling case for MAP65-3 as a primary effector of α -Aurora at the phragmoplast midzone.

*Another fact that questions significance of MAP65-3 phosphorylation by α -Aurora is differences between phenotype of *aur1aur2* mutant and line expressing MAP65-3AA. In *aur1aur2* MAP65-3 does not localize to the phragmoplast tip (Figure 5C) whereas MAP65-3AA labels the entire phragmoplast (Figure 6C). Considering this discrepancy the authors should explore other substrates of α -Aurora kinase by comparing the phosphoproteome of Col-0 and *aur1aur2*.*

Response: Actually, localization of MAP65-3 in *aur1 aur2* cells resembles MAP65-3^{AA} localization in *map65-3* cells. Both show wider localization at early stages of cytokinesis but mainly concentrate at phragmoplast midzone at late stages of cytokinesis. In Figure 6C, we aim to show that MAP65-3^{AA} is retained in the interior region of the ring-shaped phragmoplast, though it appears to also label the leading tip. This is because the cell imaged is not at a very late stage of cytokinesis, so the phragmoplast has not yet become a fully rounded ring. Our new live-cell imaging data confirms that MAP65-3^{AA} does not label the leading phragmoplast tip at late stages of cytokinesis (Figure 6H).

We agree phosphoproteomic analyses comparing wild-type and *aur1 aur2* cells would provide valuable insight into α -Aurora's diverse substrates and signaling relationships. However, such global studies require not just broad analyses but targeted validation and study of each substrate/pathway identified. This will not only take years of concerted effort but also dilute the focus of the current story if included. We aim to take a stepwise approach, first establishing discrete mechanisms like the α -Aurora-MAP65-3 signaling at the phragmoplast midzone. We want to develop a systems-level view of how α -Aurora coordinates cell division, built gradually on a foundation of solid mechanistic understanding.

*Inconsistency of experimental approaches to protein localization is a significant concern. Many localization studies are performed using chemical fixation, which is prone to artefacts. As a consequence there are several confusing results. For example, the lack of MAP65-3 localization at the phragmoplast tip in Figure 5C is a common artefact of chemical fixation. The authors should generate *aur1aur2* mutants expressing MAP65-3 and tubulin markers and analysed co-localization of MAP65-3 for these experiments. It is also important to analyse co-localization of cell plate assembly markers KNOLLE and MAP65-3 on Col-0 and *aur1aur2* backgrounds. Another example is Figure 6. The top row in panel F is labelled as the MAP65-3AA mutant whereas the localization pattern is identical to the MAP65-3DD mutant in Figure 6C. The bottom row in panel F is labelled MAP65-3DD, but the localization pattern is identical to the AA mutant in panel C. These comparisons should be performed in living cells during phragmoplast expansion.*

Response: We agree additional experiments using live-cell imaging are needed to conclusively

determine MAP65-3 localization. To address this point, we deliver a *pTUB6:mCherry-TUB6* marker to our lines and perform live-cell confocal imaging of MAP65-3-GFP and mCherry-tubulin simultaneously. Live-cell confocal imaging of these lines reveals that MAP65-3 in *aur1 aur2* background or MAP65-3^{AA} in *map65-3* background retains in the interior region of the ring-shaped phragmoplast but less decorates its leading edge (Figure 4E, Figure 6H). These new live-cell imaging results are consistent with our original immunofluorescence findings. We have also analyzed co-localization of MAP65-3 and the cell plate marker FM4-64 via live-cell imaging (Figure 5-7). Again, MAP65-3^{AA} signal predominantly labels the interior region of the mature cell plate (Figure 6J).

The original FRAP experiments for MAP65-3^{AA} shown in Figure 6F is conducted at a late phragmoplast stage, while MAP65-3^{DD} is imaged at an early stage, leading to an impression that MAP65-3^{AA} and MAP65-3^{DD} have similar localization patterns. Although the turnover rate of MAP65-3 is consistent between the early and late phases of cytokinesis (Supplemental Figure S4), we have updated the MAP65-3^{AA} FRAP figures to an earlier cytokinetic phase to avoid any confusion (Figure 6F). Based on clarification of our original results and additional live-cell imaging experiments, we believe the seemingly inconsistencies are now fully addressed.

Expansion rate in Figure 5 was measured incorrectly. As Figure 5C shows that MAP65-3 may not always follow the phragmoplast tip, the expansion rate should be measured using a microtubule marker.

Response: We appreciate the reviewer raising this astute observation and suggestion. To address this point, we have generated MAP65-3^{AA/DD} lines co-expressing mCherry-tubulin to mark microtubules in the same cells (Figure 6H). In these lines, we measured the phragmoplast expansion rate using the MTs signal. These new measurements confirm our original findings that the expansion rate of the phragmoplast is significantly decreased in the MAP65-3^{AA} line when compared to the MAP65-3 and MAP65-3^{DD} lines (Figure 6I).

The authors should demonstrate how phosphorylation affects activity of MAP65-3. Slower turnover in the midzone in Figure 6G could be a consequence of defective cell plate assembly or association with other proteins. These possibilities should be verified. As authors already have lines expressing GFP-fusions of mutants, a simple-immunoprecipitation will provide information about interactomes of the mutants.

Response: We have examined cell plate formation in the α -Aurora double mutant and found it is able to form mature cell plates. We also have attempted co-immunoprecipitation experiments from lines expressing MAP65-3-GFP to compare interactomes between wild-type and *aur1 aur2* plants. However, MAP65-3 is always degraded in these experiments, yielding insufficient bait peptides for a robust and insightful analysis. As an alternative approach, we perform yeast two-hybrid assays and find phosphorylation did not affect MAP65-3 interaction with known partners BUB3.1 and POK2 (Y2H figures above). Additionally, proteins thought to associate with MAP65-3, including BUB3.1, Kinesin-12A and TRS120, retain their phragmoplast midzone localization pattern in *aur1 aur2* cells. (Supplemental Figure S5). Together, these results suggest phosphorylation regulates the kinetics of MAP65-3 interaction with phragmoplast but does not dramatically alter MAP65-3 interactions at the phragmoplast midzone.

Regarding the slower turnover of MAP65-3^{AA} in the phragmoplast midzone, we do not think this is due to defective cell plate assembly or disruption of protein interactions. Rather, we believe the slower turnover of MAP65-3^{AA} directly is resulted from enhanced microtubule association. By comparing the turnover of MAP65-3 and MAP65-3^{AA} in tobacco epidermal cells, we find MAP65-3^{AA} exhibits a substantially decreased turnover rate (Supplemental Figure S7). Given that MAP65-3 primarily bundles interphase cortical microtubules and has less opportunity to associate with cell plate factors in the tobacco-based transient expression system, we conclude that phosphorylation by α -Aurora kinases directly regulates the microtubule-binding turnover of MAP65-3 in a cell cycle-dependent manner.

*Analysis of interaction between MAP65-3 and α -Aurora is preliminary and requires functional characterization. The authors should examine the function of interaction between α -Aurora and MAP65-3 by complementing *aur1aur2* with α -Aurora mutant that does not bind MAP65-3 and complement *map65-3* with MAP65-3 mutant that does not bind Aurora A.*

Response: For α -Aurora, it is a small kinase that contains mostly a large kinase domain with very few other regions. We have attempted but are unsuccessful in mapping the region interacting with MAP65-3, as deleting only a small region from either the N- or C-terminus in AUR1 disrupts the interaction. We propose that the interaction between α -Aurora and MAP65-3 is crucial for the phosphorylation of MAP65-3 and its subsequent functional regulation. Thus we prefer using phosphosite-mutation assays to examine MAP65-3 function. The altered localization patterns and dynamics of MAP65-3^{AA} are consistent with changes in MAP65-3 activities observed in the *aur1 aur2* mutant (Figure 6). Additionally, the phosphomimetic form of MAP65-3 conspicuously alleviates growth and localization defects observed in the α -Aurora mutant cells (Figure 7). We believe the new evidence presented sufficiently supports phospho-regulation of MAP65-3 dynamics as a key mechanism by which α -Aurora facilitates rapid phragmoplast progression.

*Association of MAP65-3 with microtubules in *aur1aur2* suggests that MAP65-3 can bind parallel microtubules. However, during cytokinesis in *aur1aur2* MAP65-3 binds only antiparallel microtubules. These data suggest that phosphorylation by α -Aurora changes microtubule bundling mechanisms by MAP65-3. This should be tested by determining polarity of microtubule bundles produced by the mutants and wild type MAP65-3 in vitro.*

Response: We suppose phosphorylation by α -Aurora affects MAP65-3 associating with microtubules rather than recognizing anti-parallel microtubule overlaps. Firstly, both phosphorylation sites locate at the microtubule binding domain but not the homodimerization domain in MAP65-3. Secondly, FRAP experiments in tobacco leaf cells show that the turnover rate of the non-phosphorylatable MAP65-3^{AA} on microtubules is slower than the wild-type MAP65-3 (Supplemental Figure S7), suggesting α -Aurora-mediated phosphorylation causes MAP65-3 to detach from microtubules. During anaphase to telophase, the anti-parallel microtubule overlapping regions in the central spindle are wide. At this stage in *aur1 aur2* cells, MAP65-3 fails to detach from central spindle microtubules in a timely manner, causing it to appear to label the entire central spindle or early phragmoplast, resembling decoration of parallel microtubules. During cytokinesis, MAP65-3 mainly decorates the midzone rather than the whole phragmoplast in *aur1 aur2* plants. This suggests that while phosphorylation by α -Aurora kinase

may stimulate the detachment of MAP65-3 from microtubules *in vivo* to properly localize it during cytokinesis, the intrinsic ability of MAP65-3 to selectively bind and link anti-parallel microtubules does not rely on Aurora-mediated phosphorylation. Our *in vitro* microtubule bundling assays with wild-type MAP65-3 and the non-phosphorylatable MAP65-3^{AA} mutant demonstrate that neither protein decorates all microtubules within the bundles. Thus, we believe MAP65-3 appears to inherently recognize anti-parallel microtubule overlaps, which does not depend on regulation by Aurora kinases.

Minor weaknesses:

Figure 1 has been published before and should be removed from the paper.

Response: We have moved Figure 1 to Supplemental Materials.

Reviewer #1 (Remarks to the Author):

The authors have addressed all the technical concerns raised by this reviewer; I believe that the manuscript is publishable in its current form. Meanwhile, I hold my initial view that the conceptual advances made in this study are marginal and the study falls short of the level of Nature Communications.

Reviewer #2 (Remarks to the Author):

This is a revised manuscript that I reviewed several months ago. In response to my comments the authors performed additional experiments and modified the text. I appreciate the efforts in providing additional experimental data and explanations.

Although this version includes a sizable dataset, papers published in Nature Communications are supposed to address mechanisms of biological processes by providing a substantial body of experimental data. In my opinion, the existing data is inconclusive mostly because my major criticism was not addressed directly though these experiments are relatively fast and simple. Additional experimental evidence is necessary to answer the question about the role of phosphorylation of MAP65-3 by Aurora.

It has been shown that MAP65-3 plays two main roles: bundling anti-parallel microtubules and targeting other proteins to the midzone. Before the manuscript is published in Nature Communications the authors should perform experiments that answer which of the functions or both are affected by the phosphorylation.

The first major concern about the dataset is that MAP65-3 still localizes to the midzone in the *aur1aur2* as in the Col-0. The only phenotype is slower turnover of MAP65-3 in the midzone. However, this phenotype is unlikely to be a consequence of interaction between MAP65-3 and microtubules because as shown in Figures 3 and 4, MAP65-3 remains in the midzone even after depolymerization of microtubules. This fact means that phosphorylation controls retention of MAP65-3 in the midzone through interaction with other proteins. The author analyzed localization of several known MAP65-3 interactors and midzone makers such as BUB3.1, Kin-12A, TRS120, and Knolle. All these proteins showed normal localization. However, there are many midzone proteins that were not tested. The authors should compare interactomes of MAP65-3, MAP65-3DD, and MAP65-3AA. This analysis will reveal which protein could be responsible for keeping MAP65-3 in the midzone after depolymerization of microtubules.

The second major concern is that the claim that MAP65-3DD "alleviates growth and localization defects observed in the α -Aurora mutant cells" is not substantiated by the data. The phenotype of this line is as severe as *aur1aur2*. It is highly likely that other pathways affect MAP65-3 activity in *aur1aur2*. Providing mechanisms of Aurora regulating MAP65-3 requires insight in these pathways. In particular, MAPK pathway was implicated in the control of MAP65-1 activity and Aurora knockout could perturb MAPK signaling. My original comments suggested comparing phosphorylation status of MAP65-3 in Col-0 and *aur1aur2* using phosphoproteomics. This experiment is critical in order to demonstrate whether other phosphorylation events on MAP65-3 are perturbed in *aur1aur2*. Many proteomics service centers can complete such an analysis within three to four weeks, which is reasonable for the revision timeline.

The third critical concern is the claim that phosphorylation by Aurora restricts activity of MAP65-3. However, the phragmoplast expands normally. Further, MAP65-3DD can rescue *map65-3* mutant phenotype but can not rescue *aur1aur2* phenotype. The data provided in the manuscript strongly supports hypothesis that MAP65-3DD mutant functions normally and phosphorylation does not affect its activity. The authors should perform direct functional assays by test the impact of MAP65-3DD on microtubule dynamics and bundling in vitro and in vivo. These data will determine whether phosphorylation restricts or increases activity of MAP65-3.

We sincerely appreciate the reviewer providing valuable feedback and insightful critiques on our initial revised manuscript. In response to these comments, we have conducted additional experiments comparing the MAP65-3 interactome and phosphoproteome between WT control and *aur1 aur2* plants. We have also made the suggested revisions to the manuscript text and figures based on the reviewer's recommendations, as detailed in our point-by-point responses below.

Reviewer #2 (Remarks to the Author):

This is a revised manuscript that I reviewed several months ago. In response to my comments the authors performed additional experiments and modified the text. I appreciate the efforts in providing additional experimental data and explanations.

Although this version includes a sizable dataset, papers published in Nature Communications are supposed to address mechanisms of biological processes by providing a substantial body of experimental data. In my opinion, the existing data is inconclusive mostly because my major criticism was not addressed directly though these experimental are relatively fast and simple. Additional experimental evidence is necessary to answer the question about the role of phosphorylation of MAP65-3 by Aurora.

It has been shown that MAP65-3 plays two main roles: bundling anti-parallel microtubules and targeting other proteins to the midzone. Before the manuscript is published in Nature Communications the authors should perform experiments that answer which of the functions or both are affected by the phosphorylation.

*The first major concern about the dataset is that MAP65-3 still localizes to the midzone in the *aur1aur2* as in the Col-0. The only phenotype is slower turnover of MAP65-3 in the midzone. However, this phenotype is unlikely to be a consequence of interaction between MAP65-3 and microtubules because as shown in Figures 3 and 4, MAP65-3 remains in the midzone even after depolymerization of microtubules. This fact means that phosphorylation controls retention of MAP65-3 in the midzone through interaction with other proteins. The author analyzed localization of several known MAP65-3 interactors and midzone makers such as BUB3.1, Kin-12A, TRS120, and Knolle. All these proteins showed normal localization. However, there are many midzone proteins that were not tested. The authors should compare interactomes of MAP65-3, MAP65-3DD, and MAP65-3AA. This analysis will reveal which protein could be responsible for keeping MAP65-3 in the midzone after depolymerization of microtubules.*

Response: We want to clarify that the retention of MAP65-3 is always coincident with residual microtubules at the phragmoplast midzone in *aur1 aur2* cells, based on our observations. At intermediate stages of cytokinesis, these incompletely depolymerized microtubules are difficult to discern due to the overwhelming fluorescence signal from the high density of microtubules in neighboring areas. However, low amounts of undepolymerized microtubules are present in the interior region when the phragmoplast first transitions to a ring shape. We apologize for the confusion due to not capturing later cytokinesis stages showing full microtubule disassembly in our main figures. As shown in Movie S10, once phragmoplast microtubules are completely depolymerized at late cytokinesis (over 30 min of cytokinesis), almost all MAP65-3 signal no longer remains at the interior phragmoplast midzone in *aur1 aur2* cells (see below figure). The delayed MAP65-3 turnover we observe in *aur1 aur2* cells is coincident with residual microtubules that fail to fully disassemble. We propose the slower detachment of MAP65-3 from these

remaining microtubules is what restricts their depolymerization and phragmoplast expansion.

Previously, we attempted co-immunoprecipitation from transgenic MAP65-3-GFP floral tissue but identified only 6 peptides of the bait protein by mass spectrometry, without recovery of any known phragmoplast midzone, microtubule associated, or putative interaction partners (Supplemental Table S3). The unsuccessful tries prompted us to try semi-in vivo immunoprecipitation approaches using recombinant GST-MAP65-3 immobilized on glutathione beads and protein preparations from WT or *aur1 aur2* flower buds. This approach yielded a substantial amount of bait protein and co-precipitated candidates (see below table summarized from Supplemental Table S4), including known phragmoplast midzone factors like DRP1s, kinesin-12, BUB3, and myosins, along with other microtubule-associated proteins. However, a general comparison revealed no significant differences in the MAP65-3 interaction network between WT and *aur1 aur2* plants. Thus, our proteomics data did not uncover changes to the MAP65-3 interactome aside from microtubule binding effects that could explain its retention in *aur1 aur2* cells.

Hits			Baits			
Protein ID	Gene ID	Name	MAP65-3 from WT		MAP65-3 from aur1 aur2	
			Peptides (n)	Coverage	Peptides (n)	Coverage
Q9FHM4	AT5G51600	MAP65-3	74	73%	85	77%
P42697	AT5G42080	DRP1A	23	43%	18	33%
Q8LF21	AT1G14830	DRP1C	10	17%	6	11%
Q9FNX5	AT3G60190	DRP1E	8	14%	10	17%
P41916	AT5G20010	RAN-1	6	28%	5	22%
Q9LZY0	AT3G60840	MAP65-4	5	10%	6	10%
F4J464	AT3G23670	Kinesin-12B	3	5%	3	4%
Q9C701	AT1G49910	BUB3.2	2	4%	2	5%
Q39024	AT4G01370	MPK4	2	5%	3	7%
Q39160	AT1G17580	Myosin XI-1	2	3%	3	3%
F4JCF9	AT3G19960	Myosin 1	1	1%	2	2%
A0A1P8ARX4	AT1G04160	Myosin XI-8	1	1%	2	4%
A0A1P8BH66	AT5G60210	RIP5	1	2%	2	3%
Q9LEZ4	AT5G08120	MPB2C	3	10%	1	3%
P24100	AT3G48750	CDKA;1	2	6%	1	2%
Q8L4D8	AT1G74690	IQD31	1	2%	1	2%
P46864	AT4G27180	ATK2	1	1%	1	1%
Q9LX99	AT5G10470	KAC1	1	1%	2	2%
Q940Y8	AT3G16060	Kinesin-13B	1	2%	1	2%
F4I061	AT1G49040	SCD1	1	1%	1	1%
Q9T041	AT4G27060	TOR1	1	1%	1	1%

The second major concern is that the claim that MAP65-3^{DD} “alleviates growth and localization defects observed in the α -Aurora mutant cells” is not substantiated by the data. The phenotype of this line is as severe as *aur1 aur2*. It is highly likely that other pathways affect MAP65-3 activity in *aur1 aur2*. Providing mechanisms of Aurora regulating MAP65-3 requires insight in these pathways. In particular, MAPK pathway was implicated in the control of MAP65-1 activity and Aurora knockout could perturb MAPK signaling. My original comments suggested comparing phosphorylation status of MAP65-3 in Col-0 and *aur1 aur2* using phosphoproteomics. This experiment is critical in order to demonstrate whether other phosphorylation events on MAP65-3 are perturbed in *aur1 aur2*. Many proteomics service centers can complete such an analysis within three to four weeks, which is reasonable for the revision timeline.

Response: As noted, α -Aurora is known to be a key regulator of spindle assembly during mitosis. The dwarf phenotype observed in *aur1 aur2* plants likely stems primarily from defects in spindle function. Since MAP65-3 acts at the phragmoplast midzone rather than on spindles. Thus, expressing MAP65-3^{DD} cannot fully rescue the *aur1 aur2* mutant defects caused by impaired spindle activity. However, we did observe partial growth improvement and proper localization of MAP65-3^{DD} when expressed in the *aur1 aur2* background. This suggests α -Aurora may contribute to a subset of MAP65-3 phosphorylation events during cytokinesis, though spindle regulation is its major role. In the revised manuscript, we have pointed out MAP65-3^{DD} partially suppressed the busy growth phenotype of *aur1 aur2* plants.

MAP65-3 from WT background		MAP65-3 from aur1 aur2 background	
Peptide	Residue	Peptide	Residue
⁶⁸ QAIADAEQLAAIC S AMGERPVHIR ⁹² ($\times 7$)	S82	⁶⁸ QAIADAEQLAAIC S AMGERPVHIR ⁹² ($\times 10$)	S82
⁹³ QSDQSVG S LKQELGR ¹⁰⁷ ($\times 2$)	S94	⁶⁸ QAIADAEQLAAIC S AMGERPVHIR QSDQSVG SLK ¹⁰² ($\times 2$)	S94
⁹³ QSDQSVG S LK ¹⁰² ($\times 1$) ⁹³ QSDQSVG S LKQELGR ¹⁰⁷ ($\times 2$)	S100	⁶⁸ QAIADAEQLAAIC S AMGERPVHIR QSDQSVG SLK ¹⁰² ($\times 1$) ⁶⁸ QAIADAEQLAAIC S AMGERPVHIR QSDQSVG SLKQELGR ¹⁰⁷ ($\times 1$)	S100
¹⁴² GQGELVHSEPLID E TNLSMRK ¹⁶² ($\times 1$) ¹⁴² GQGELVHSEPLID E TNLSMR ¹⁶¹ ($\times 1$)	T156	¹²¹ RNQFIVVMEQID S ITNDIKGQGELVHSEPLID E TNLSMR ¹⁶¹ ($\times 1$) ¹⁴² GQGELVHSEPLID E TNLSMR ¹⁶¹ ($\times 1$)	T156
⁴⁶⁰ LL S MLEEYNILRQEREEHR ⁴⁷⁹ ($\times 1$) ⁴⁶⁰ LL S MLEEYNILRQER ⁴⁷⁴ ($\times 2$)	S462	⁴⁶⁰ LL S MLEEYNILR ⁴⁷¹ ($\times 1$) ⁴⁶⁰ LL S MLEEYNILRQER ⁴⁷⁴ ($\times 1$)	S462
⁴⁸⁶ KLQGQLIAEQEAL Y GSKPSPSKPLGGK ⁵¹² ($\times 2$)	Y499	⁴⁸⁶ KLQGQLIAEQEAL Y GSKPSPSKPLGGK ⁵¹² ($\times 1$) ⁴⁸⁷ LQGQLIAEQEAL Y GSKPSPSKPLGGK ⁵¹² ($\times 1$)	Y499
⁴⁸⁶ KLQGQLIAEQEAL Y GSKPSPSKPLGGK ⁵¹² ($\times 2$)	S501	⁴⁸⁶ KLQGQLIAEQEAL Y GSKPSPSKPLGGK ⁵¹² ($\times 3$)	S501
⁴⁸⁷ LQGQLIAEQEAL Y GSKPSPSKPLGGK ⁵¹² ($\times 3$)	S504	⁴⁸⁶ KLQGQLIAEQEAL Y GSKPSPSKPLGGK ⁵¹² ($\times 3$)	S504
⁴⁸⁷ LQGQLIAEQEAL Y GSKPSPSKPLGGK ⁵¹² ($\times 2$)	S506	⁴⁸⁷ LQGQLIAEQEAL Y GSKPSPSKPLGGK ⁵¹² ($\times 1$) ⁴⁸⁶ KLQGQLIAEQEAL Y GSKPSPSKPLGGK ⁵¹³ ($\times 1$)	S506
⁵²⁶ RL S LGAAAMHQT P KPNK ⁵⁴¹ ($\times 3$) ⁵²⁷ LSLGAAAMHQT P KPNK ⁵⁴¹ ($\times 2$)	S528	⁵²⁷ LSLGAAAMHQT P KPNK ⁵⁴² ($\times 2$)	T536
⁵²⁷ LSLGAAAMHQT P KPNK ⁵⁴² ($\times 1$) ⁵²⁶ RL S LGAAAMHQT P KPNK ⁵⁴¹ ($\times 1$)	T536		
⁵⁶⁸ KQ S MNPSEMLQ S PLVR ⁵⁸³ ($\times 7$) ⁵⁶⁹ Q S MNPSEMLQ S PLVR ⁵⁸³ ($\times 3$)	S570		

We apologize for initially misunderstanding the suggestions to perform phosphoproteomics analysis. The reviewer makes an excellent point that disrupting α -Aurora could potentially alter MAP65-3 phosphorylation by other kinases. To directly test this, we re-examined the MAP65-3 phosphoproteome from wild-type and *aur1 aur2* plants using semi-in vivo immunoprecipitation and mass spectrometry. We

identified similar MAP65-3 phosphosites in both genotypes, except for loss of the two α -Aurora-targeted residues S528 and S570 in *aur1 aur2* plants (see above table summarized from Supplemental Table S5). This result indicates no dramatic perturbation of MAP65-3 phosphorylation by other kinases when α -Aurora is disrupted. We apologize for not conducting this analysis at our first revision, and thank the reviewer for this insightful suggestion to strengthen our study.

The third critical concern is the claim that phosphorylation by Aurora restricts activity of MAP65-3. However, the phragmoplast expands normally. Further, MAP65-3DD can rescue map65-3 mutant phenotype but cannot rescue aur1 aur2 phenotype. The data provided in the manuscript strongly supports hypothesis that MAP65-3DD mutant functions normally and phosphorylation does not affect its activity. The authors should perform direct functional assays by test the impact of MAP65-3DD on microtubule dynamics and bundling in vitro and in vivo. These data will determine whether phosphorylation restricts or increases activity of MAP65-3.

Response: We apologize for the confusion caused by our claim that Aurora phosphorylation restricts MAP65-3 “activity”. To clarify, we meant that α -Aurora-mediated phosphorylation promotes detachment of MAP65-3 from microtubules, thereby facilitating its turnover at the phragmoplast midzone. In other words, phosphorylation by α -Aurora restricts the extent of MAP65-3's microtubule binding. Previous studies have shown that phosphorylation-dependent reduction of MAP65 microtubule binding activity enables microtubule turnover and phragmoplast expansion during cytokinesis (Smertenko et al., 2006, J Cell Sci, 119:3227-3237; Sasabe et al., 2006, Genes Dev, 20:1004-1014).

As suggested, we directly examined whether α -Aurora phosphorylation impacts MAP65-3's microtubule binding ability. As shown in the new Supplemental Figure S7 (also attached below), FRAP experiments in tobacco epidermal cells revealed the non-phosphorylatable MAP65-3^{AA} detached from microtubules more slowly compared to wild-type MAP65-3, exhibiting delayed fluorescence recovery. In contrast, the phosphomimetic MAP65-3^{DD} turnover was enhanced, with more rapid signal recovery. The slower and faster turnover rates of MAP65-3^{AA} and MAP65-3^{DD}, respectively, further demonstrate that α -Aurora-mediated phosphorylation facilitates dissociation of MAP65-3 from microtubules.

Our objective was to demonstrate that α -Aurora-mediated phosphorylation modulates MAP65-3's association with microtubules at the phragmoplast midzone during cytokinesis. However, we should not have described this as restricting MAP65-3 “activity”, which implies effects on a defined biochemical function. We have removed vague references to restricted “activity” from the text and now more precisely describe the effects of phosphorylation on MAP65-3's microtubule binding capacity, turnover dynamics, and localization during cytokinesis. We have also modified the title to “Arabidopsis α -Aurora Kinase Plays a Role in Cytokinesis through Regulating MAP65-3 Association with Microtubules at Phragmoplast Midzone” to better convey this concept. We thank the reviewer for catching this imprecise wording, and have revised the manuscript to more accurately convey the impact of Aurora-dependent phosphorylation on MAP65-3's microtubule association without implying other functional activities.

Reviewer #2 (Remarks to the Author):

The revised manuscript includes three novel datasets: (i) phosphorylated residues of MAP65-3 in *aur1aur2* background and in Col-0 control; (ii), interactome of MAP65-3 in *aur1aur2* and in Col0 control; and (iii) turnover rate of MAP65-3, MAP65-3AA and MAP65-3DD in the interphase cells. I much appreciate the efforts to produce these datasets. The authors use this evidence to support the hypothesis that Aurora kinase regulates cytokinesis by reducing affinity between MAP65-3 and microtubules in the midzone.

My major concern is that numerous experimental data contradict this conclusion.

1. The phragmoplast looks normal in *aur1aur2* in Figure 3D, Figure 4C,E, Figure 5B, and Supplemental Figure 5A-C. Somewhat wider phragmoplast midzone in Figure 1B could be a natural variability of the phragmoplast morphology. In fact the midzone of the phragmoplast in Col-0 in Figure 1A seems wider in the time frames 4:00 and 13:40. The phragmoplast branching phenotype is only shown in Figure 5. The lower frequency of the phenotype questions whether Aurora kinase contributes to regulation of phragmoplast morphology.
2. Figure 4C and E show similar localization of MAP65-3 in Col-0 and *aur1aur2*. This outcome is inconsistent with the idea that phosphorylation of MAP65-3 by aurora controls phragmoplast expansion.
3. Figure 4D shows localization of MAP65-3 at the central part of the phragmoplast lacking microtubules. Figure 4E 12' time frame also shows MAP65-3 in the region devoid of microtubules. In the response to my comment, the authors claim that this region contains remnant microtubules, these microtubules are invisible on the images provided though. If these regions contain microtubules, then it is important to provide relevant supporting information. Unless this data is provided, there is still a possibility that the phosphorylation controls interaction of MAP65-3 with other proteins or phosphorylation of other proteins determines localization of MAP65-3.
4. Localization of MAP65-3 interacting proteins is affected in *aur1aur2*. Both BUB3.1 and Kin12-A localize in the region devoid of microtubules (Supplemental Figure S5). This outcome provides a very strong support that MAP65-3 is retained in the midzone after microtubule depolymerization due to interaction with BUB3.1, Kin12A, or both, but not because of stronger affinity with microtubules.
5. If constitutively active MAP65-3 was responsible for the cytokinetic defects in *aur1aur2*, then the ectopic expression of MAP65-3AA would cause similar defects. However, phragmoplast morphology was not affected in cells expressing MAP65-3AA as shown in Figure 6H.

Reviewer #2 (Remarks to the Author):

*The revised manuscript includes three novel datasets: (i) phosphorylated residues of MAP65-3 in *aur1aur2* background and in *Col-0* control; (ii), interactome of MAP65-3 in *aur1aur2* and in *Col-0* control; and (iii) turnover rate of MAP65-3, MAP65-3AA and MAP65-3DD in the interphase cells. I much appreciate the efforts to produce these datasets. The authors use this evidence to support the hypothesis that Aurora kinase regulates cytokinesis by reducing affinity between MAP65-3 and microtubules in the midzone.*

My major concern is that numerous experimental data contradict this conclusion.

*1. The phragmoplast looks normal in *aur1aur2* in Figure 3D, Figure 4C, E, Figure 5B, and Supplemental Figure 5A-C. Somewhat wider phragmoplast midzone in Figure 1B could be a natural variability of the phragmoplast morphology. In fact the midzone of the phragmoplast in *Col-0* in Figure 1A seems wider in the time frames 4:00 and 13:40. The phragmoplast branching phenotype is only shown in Figure 5. The lower frequency of the phenotype questions whether Aurora kinase contributes to regulation of phragmoplast morphology.*

Response: We wish to clarify that we did not aim to propose a severely disrupted or disorganized phragmoplast structure in *aur1 aur2* plants. Rather, our results seek to demonstrate that the process of phragmoplast expansion appears obstructed or hindered in the mutant compared to WT plants. Upon those mentioned figures, we believe the key conclusions supported by our results are: (1) Phragmoplast expansion is significantly slower in *aur1 aur2* compared to WT based on live-cell imaging analysis in Figure 1; (2) MAP65-3 displays altered dynamics at phragmoplast microtubule arrays in *aur1 aur2* cells, with preferential accumulation at trailing edges and persistent association at dismantling interiors seen in Figures 3-5. The wider midzone and disorganized leading edges are often observed when phragmoplast expands to a matured ring-shape stage, rather than as a general defect. We have already pointed these observations when mentioned Figure 1 in the Results part. Furthermore, as shown in Figure 1 and Supplementary Movie 1, quantification of the phragmoplast midzone width over time demonstrate that, unlike in *aur1 aur2* cells, the gap width remains largely unchanged in WT cells as the phragmoplast expanding centrifugally.

In Figure 5, our objective is to demonstrate the consistent enrichment and persistence of MAP65-3 signal specifically at the branch zones of a mature cell plate within *aur1 aur2* plants. We also asserted that only a small subset of *aur1 aur2* cells exhibited branched cell plates, and discussed the interplay between α -Aurora and membrane remodeling in our previous revision. It's important to note that the main aim of Figure 5 is not to over-interpret this less frequent branching phenotype, but rather to showcase how the turnover dynamics of MAP65-3 are altered during cytokinesis, resulting in its preferential accumulation at the interiors of mature cell plates.

Upon re-evaluation, we have revised the conclusions to tone down any implications of strong defects or direct control of phragmoplast morphology, and instead focus on the role of α -Aurora in facilitating phragmoplast expansion through phosphorylation-dependent effects on MAP65-3 turnover and binding with anti-parallel microtubules at the phragmoplast midzone.

*2. Figure 4C and E show similar localization of MAP65-3 in *Col-0* and *aur1aur2*. This outcome is inconsistent with the idea that phosphorylation of MAP65-3 by *aurora* controls phragmoplast expansion.*

Response: Regarding Figure 4C and E, the MAP65-3 localization patterns are not in fact similar between control and *aur1 aur2* plants:

- In control cells during late cytokinesis when the phragmoplast is dismantling at the center, MAP65-3 cleanly removes from the interior region where shows invisible microtubule fluorescence.
- By contrast, in *aur1 aur2* cells MAP65-3 persistently associates within the dismantling interior zone, indicating it fails to timely disengage from the remnant phragmoplast structure.
- Additionally, at earlier timepoints MAP65-3 decorates microtubule edges more uniformly across the phragmoplast midzone in control cells, whereas in *aur1 aur2* cells it abnormally accumulates in trailing edges of the ring-shaped phragmoplast.

The distinct variances in MAP65-3 distributions between the two lines corroborate our model, suggesting that altered phosphorylation affects its timely turnover during phragmoplast expansion.

3. Figure 4D shows localization of MAP65-3 at the central part of the phragmoplast lacking microtubules. Figure 4E 12' time frame also shows MAP65-3 in the region devoid of microtubules. In the response to my comment, the authors claim that this region contains remnant microtubules, these microtubules are invisible on the images provided though. If these regions contain microtubules, then it is important to provide relevant supporting information. Unless this data is provided, there is still a possibility that the phosphorylation controls interaction of MAP65-3 with other proteins or phosphorylation of other proteins determines localization of MAP65-3.

Response: We apologize for any confusion caused by implying MAP65-3 is permanently localized in microtubule-free regions. Upon re-examining Movie S10 over a longer timeframe of phragmoplast expansion (over 30 minutes), the MAP65-3 signal does eventually disengagement from the central region in *aur1 aur2* cells. However, this process is delayed compared to control cells, demonstrating slower turnover rates. The intent of Figure 4 was to illustrate these kinetic delays. Additionally, contrary to the reviewer's observation, residual microtubules at the phragmoplast center are indeed visible at the 12' timepoint in Figure 4E (arrows), indicating a stalled rather than fully absent maturation of the ring-shaped structure in *aur1 aur2* cells. The delayed maturation of the phragmoplast in *aur1 aur2* cells is likely due to the slowed movement of MAP65-3 out of the central region toward the leading edges.

It should also be noted that the interdigitated microtubules at the phragmoplast midzone where MAP65-3 localized appears as an invisible dark line under both live-cell fluorescence imaging and immunofluorescence microscopy. It is known that the phragmoplast midzone region contains a significantly smaller number of interdigitated microtubules compared to the periphery of the phragmoplast. This sparse density of residual midzone microtubules makes them relatively faint and difficult to discern visually, thereby imparting a dark line appearance. We suppose there are likely still residual interdigitated microtubules present that fail to dismantle timely caused by impaired MAP65-3 turnover kinetics in *aur1 aur2* cells, even as the bulk of microtubules depolymerize radially at the distal phragmoplast zone.

Our earlier proteomics experiments did not find any significant changes in MAP65-3 interacting proteins regulated by Aurora phosphorylation, but we acknowledge it is challenging to thoroughly screen for proteins specifically expressed during cell division using current techniques. Because such proteins may only be present in low amounts limited to mitotic/cytokinetic stages, rather than throughout the cell cycle, they can be difficult to detect. A thorough investigation of additional phosphorylation targets and binding partners that may contribute to MAP65-3 localization is certainly warranted but beyond the intended scope of the present study. As the reviewer insightfully points out, we cannot definitively rule out the possibility that MAP65-3 phosphorylation may also influence its interactions with other binding partners besides microtubules. In our revised discussion section, we have pointed out that MAP65-3 phosphorylation could also modify its associations with other proteins, which may contribute to the observed phenotypes and deserves further study.

4. Localization of MAP65-3 interacting proteins is affected in aur1aur2. Both BUB3.1 and Kin12-A localize in the region devoid of microtubules (Supplemental Figure S5). This outcome provides a very strong support that MAP65-3 is retained in the midzone after microtubule depolymerization due to interaction with BUB3.1, Kin12A, or both, but not because of stronger affinity with microtubules.

Response: As we have previously reported, the phragmoplast localization of both BUB3.1 and Kin12A is known to be dependent on MAP65-3 recruitment. Given this established relationship, it logically follows that when the behavior of MAP65-3 is aberrantly altered in *aur1 aur2* plants, as our data demonstrates, the localization patterns of BUB3.1 and Kin12A would similarly exhibit changes simply by following the dynamics of their recruiting factor MAP65-3. Moreover, in our initial response file, we showed that α -Aurora phosphorylation did not impact the ability of MAP65-3 to interact with either BUB3.1 or Kin12s.

5. If constitutively active MAP65-3 was responsible for the cytokinetic defects in aur1aur2, then the ectopic expression of MAP65-3AA would cause similar defects. However, phragmoplast morphology was not affected in cells expressing MAP65-3AA as shown in Figure 6H.

Response: Again, we want to emphasize that our manuscript does not assert a severe defect in the phragmoplast structure caused by compromised α -Aurora. However, a closer look at the time-lapse data in Figure 6H does reveal a subtle phenotype of prolonged phragmoplast microtubule arrays in the 18:20 time frames of MAP65-3^{AA}-expressing cells compared to earlier time points. This phenocopies what is observed in *aur1 aur2* plants. More importantly, expressing MAP65-3^{AA} in *map65-3* cells causes similar delayed phragmoplast expansion effects as seen in *aur1 aur2* cells (Figure 6I). Additionally, the microtubule distribution patterns and turnover dynamics of MAP65-3^{AA} once again mimic what is observed in the *aur1*

aur2 mutant background. (Figure 6C, F).

Reviewer #2 (Remarks to the Author):

This is the fourth version of the manuscript by Deng et al. The textual changes to this version have increased the clarity and my concerns were addressed. Overall, the manuscript was significantly improved since the first submission.